# Accurate prediction of protein function using statistics-informed graph networks

Yaan J. Jang [1,2,7] ✉, Qi-Qi Qin[2,3,7], Si-Yu Huang[2,4,5], Arun T. John Peter[6], Xue-Ming Ding[3] & Benoît Kornmann [1] ✉

Understanding protein function is pivotal in comprehending the intricate mechanisms that underlie many crucial biological activities, with far-reaching implications in the fields of medicine, biotechnology, and drug development. However, more than 200 million proteins remain uncharacterized, and computational efforts heavily rely on protein structural information to predict annotations of varying quality. Here, we present a method that utilizes statistics-informed graph networks to predict protein functions solely from its sequence. Our method inherently characterizes evolutionary signatures, allowing for a quantitative assessment of the significance of residues that carry out specific functions. PhiGnet not only demonstrates superior performance compared to alternative approaches but also narrows the sequence-function gap, even in the absence of structural information. Our findings indicate that applying deep learning to evolutionary data can highlight functional sites at the residue level, providing valuable support for interpreting both existing properties and new functionalities of proteins in research and biomedicine.

Proteins bind to other molecules to facilitate nearly all essential biological activities. Consequently, understanding protein function is of paramount importance for comprehending health, disease, evolution, and the functioning of living organisms at the molecular level[1–3]. The primary sequence of a protein contains all the essential information required to fold up into a particular three-dimensional shape, thereby determining its activities within cells [4,5]. The evolutionary information in massive protein sequences that are gleaned from extensive genome sequencing efforts has significantly contributed to recent advances in protein structure prediction[6–9]. This evolutionary data, especially the couplings between pairwise residues, has also been utilized to characterize protein functional sites[10,11]. The evolutionary couplings have been utilized to pinpoint functional sites in proteins, capturing interactions between residues that contribute to specific functions[5,12]. Indeed, the analysis of evolutionary information has allowed the identification of allosteric mechanisms in proteins[13,14], disease variants[15], and metamorphism in proteins that undergo reversible switches between distinct folds, often accompanied by different functions[16].

To date, more than 356 million proteins in the UniProt database[17] (6/2023) have been sequenced and the vast majority (~80%) of these have no known functional annotations (e.g., enzyme commission numbers and gene ontology terms). Classical methods for annotating protein functions have been constrained by the extensive sizes of sequences, and the majority of function annotations are assigned at the protein level rather than the residue level[18,19]. As an alternative to these classical methods, computational approaches have been utilized to assign function annotations to proteins[20–24]. Notably, deep learning methods have attained remarkable accuracy in predicting protein 3D structures, surpassing the capabilities of classical approaches such as ab initio methods and homology modeling. These methods involve millions of parameters and operate without making any assumptions about the relationship between input and output data samples (e.g., AlphaFold[8] and RoseTTAFold[9]). Unlike the classical approaches, deep

[1]Department of Biochemistry, University of Oxford, Oxford, UK. [2]AmoAi Technologies, Oxford, UK. [3]School of Optical-Electrical and Computer Engineering, University of Shanghai for Science and Technology, Shanghai, China. [4]Oxford Martin School, University of Oxford, Oxford, UK. [5]School of Systems Science, Beijing Normal University, Beijing, China. [6]Institute of Biochemistry, ETH Zürich, Zürich, Switzerland. [7]These authors contributed equally: Yaan J. Jang, Qi-Qi Qin. ✉e-mail: yaan.jang@gmail.com; benoit.kornmann@bioch.ox.ac.uk

learning-based methods learn a large amount of parameters directly through the training of neural networks on extensive datasets. This enables them to generate accurate mappings from input data to expected outputs. Yet accurately assigning function annotations to proteins remains challenging, especially in comparison to experimental determinations. While there is abundant data available–whether from a single amino acid sequence, alignments of numerous homologous sequences, or protein structural information–to train deep learning-based methods, achieving accurate protein function prediction remains a persistent challenge[20–25]. Integrating physics-based knowledge from provided datasets, physics-informed deep learning methods have driven recent advances across diverse fields[26]. As a promising alternative to classical and pure deep learning techniques, they enhance the capacity of machine learning to construct interpretable methods for scientific problems. Despite decades of dedicated effort, assigning a function to a protein is more arduous than predicting its 3D structure[21,27–30]. The state-of-the-art approaches that utilize structural information have encountered less success in accurately assigning protein functions[21]. This is largely attributed to the scarcity of experimentally determined protein structures in comparison to the abundance of available sequences. Moreover, computationally predicted structures vary in their confidence scores and may not always be reliable for estimating protein function annotations, leading to variable accuracy in function annotation[21,30]. Furthermore, assessing the significance of residues using a scoring function that reliably measures their contributions to

function remains challenging, as a quantitative characterization of residue roles is not yet fully comprehended.

To address these challenges, we hypothesized that the information encapsulated in coevolving residues can be leveraged to annotate functions at the residue level. Here, we devised a statistics-informed learning approach, termed PhiGnet, to facilitate the functional annotation of proteins and the identification of functional sites. Our method capitalizes on the knowledge derived from evolutionary data to drive two stacked graph convolutional networks. Empowered by the acquired knowledge and designed network architecture, the present method can accurately assign function annotations to proteins and, importantly, quantify the significance of each individual residue with respect to specific functions.

## Results

### PhiGnet for protein function annotations

In this study, we developed the PhiGnet method using statistics-informed graph networks to annotate protein functions and to identify functional sites across species based on their sequences (Fig. 1). To assimilate knowledge from the evolutionary couplings (EVCs, relationships between pairwise residues at two co-variant sites) and the residue communities (RCs, hierarchical interactions among residues)[12], we devised the method with a dual-channel architecture, adopting stacked graph convolutional networks (GCNs) (Fig. 1a). This method specializes in assigning functional annotations, including Enzyme Commission (EC) numbers and Gene Ontology (GO) terms (biological

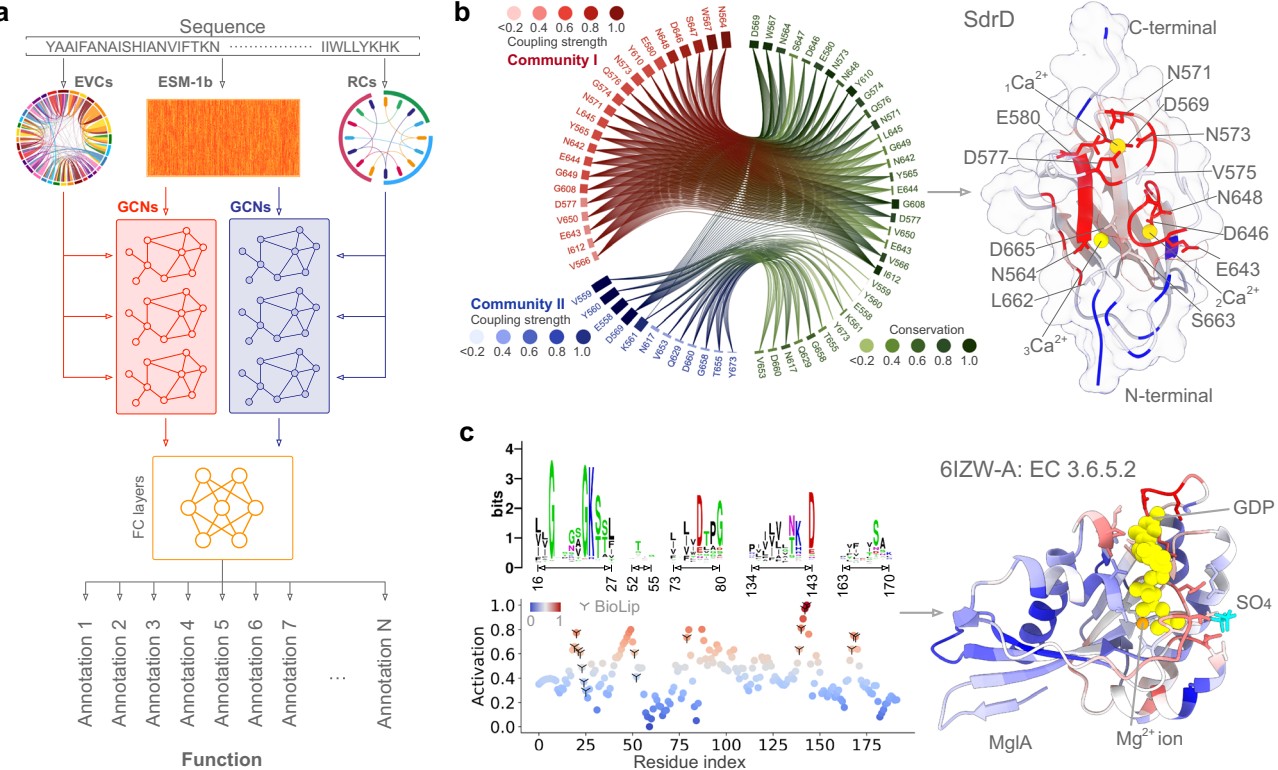

**Fig. 1 | PhiGnet annotates protein functions. a** PhiGnet predicts protein function from sequence alone. Given a sequence, PhiGnet learns the pre-embedding, EVCs, and RCs using stacked GCNs to infer protein function annotations. **b** RCs of the Serine-aspartate repeat-containing protein D (SdrD, PDB ID: 4JDZ). The two communities (community I and community II) with coupling strengths in bars are highlighted in red and blue. Each bar in either community I or II illustrates the strength of coupling that a residue has with others, while the conservation scores of these residues are depicted in the bars on the right. On the tertiary structure of SdrD (right), the residues within the community I (red) bind to the calcium ions (sphere in yellow) are shown in sticks, while the residues within the community II

(blue) adopt cartoon in blue. **c** Function annotations of the MgIA protein at the residue level. The activation score (bottom) computed by PhiGnet is to measure the importance of each residue, where the higher the score is, the more likely it is to adopt a functional role in biological activity. Compared to functional sites in BioLip (marked with Y in black), the score indicates that co-evolved residues may be more important than those at conserved positions (top). The scores are mapped to color the MgIA 3D structure (PDB ID: 6IZW) from lower (blue) to higher (red), GDP is shown with sphere in yellow, $SO_4$ in stick in cyan, and $Mg^{2+}$ ion in a sphere with orange. Source data are provided as a Source Data file.

process, BP, cellular component, CC, and molecular function, MF), to proteins. When provided with a protein sequence, we derive its embedding using the pre-trained ESM-1b model[31]. Subsequently, we input the embedding as graph nodes, accompanied by EVCs and RCs (graph edges), into the six graph convolutional layers of the dual stacked GCNs. These layers, working in conjunction with a block of two fully connected (FC) layers, meticulously process the information from the two GCNs, ultimately generating a tensor of probabilities for assessing the viability of assigning functional annotations to the protein. In addition, an activation score, derived using the gradient-weighted class activation maps (Grad-CAMs) approach[32], is used to assess the significance of each individual residue in a specific function. The score allows PhiGnet to pinpoint functional sites at the level of individual residues (bottom, Fig. 1c, see Methods).

As an example, we computed RCs for the Serine-aspartate repeat-containing protein D (SdrD) that promotes bacterial survival in human blood by inhibiting innate immune-mediated bacterial killing[33,34]. Two RCs are mapped on a fully $\beta$ sheet fold that binds to three $Ca^{2+}$ ions ($_1Ca^{2+}$ is enclosed in a loop, $_2Ca^{2+}$ is more solvent exposed and closer to $_3Ca^{2+}$, which is coordinated by an asparagine (N564) and an aspartic acid (D665), Fig. 1b). Within the community I, most residues (in red sticks) that are identified from EVCs bind to the three $Ca^{2+}$ ions, contributing together to stabilize the SdrD fold. This suggests that EVCs contain the essential information for deducing the functional role of residues, even when they are sparsely distributed across RCs. Empowered by EVCs and RCs, we implemented the present PhiGnet to assess the functional significance of residues. We carried out PhiGnet to calculate the activation scores for the functional sites of the mutual gliding-motility (MgIA) protein (annotated with EC 3.6.5.2) (Fig. 1c). The resulting activation scores show that the residues with high scores ($\geqslant 0.5$) are in agreement with or close to that of semi-manually curated BioLip database[35]. Moreover, these residues are located at the most conserved positions (top left, Fig. 1c). Upon mapping these scores onto the 3D structure of MgIA, the activation scores highlight residues (red) that constitute a pocket that binds the guanosine di-nucleotide (GDP) and play a role in facilitating nucleotide exchange[36]. Together, this suggests that residues at functional sites are conserved through natural evolution, and that PhiGnet is capable of capturing such information, improving the method for predicting protein function at the residue level, even in the absence of structural data.

## PhiGnet annotates protein functional sites
Many proteins perform their biological functions through essential residues that are sparsely distributed across different structural levels (e.g., primary, secondary, and tertiary) and are linked to functional sites (such as enzyme active sites, ligand-binding sites, or protein-protein interaction sites). Given the functional contributions of amino acids can significantly differ from one function to another, a key feature of PhiGnet is its ability to quantitatively estimate the importance of individual amino acids for a specific function, enabling us to identify residues that are pertinent to distinct biological activities.

Are the computational predictions as accurate as experimentally determined function annotations? To address this question, we carried out quantitative examinations of the contribution of each amino acid to a protein function using the activation score. We evaluated the predictive performance of PhiGnet and assessed the importance of residues (their contributions to protein function) in nine proteins: the c2-domain of cytosolic phospholipase $A_2\alpha$ (cPLA$_2\alpha$), Tyrosine-protein kinase BTK (TpK-BTK), Ribokinase, alpha-lactalbumin ($\alpha$LA), MCM1 transcriptional regular (MCM1-TR), the Fos-Jun heterodimer (FosJun), the thymidylate kinase (TmpK), Ecl18kI, and helicobacter pylori uridylate kinase (HPUK). These proteins vary in size from approximately 60 to 320 residues, harbor different folds, and perform diverse functions, including ligand binding, ion interaction, and DNA binding. We calculated the activation score for each residue in the nine proteins,

comparing them to residues identified through either experimental or semi-manual annotations. Our method demonstrated promising accuracy (with an average $\geqslant 75\%$) in predicting significant sites at the residue level, in a good agreement with actual ligand-/ion-/DNA-binding sites (Fig. 2). The activation score per residue, mapped onto their 3D structures, exhibits significant enrichment for functional relevance at the binding interfaces. PhiGnet accurately identifies functionally significant residues with high activation scores for the proteins (Fig. 2, Supplementary Figs. S1 and S2).

Across the proteins cPLA$_2\alpha$, Ribokinase, $\alpha$LA, TmpK, and Ecl18kI, PhiGnet predicted near-perfect functional sites compared to the experimental identifications. For instance, for cPLA$_2\alpha$, our method accurately identified residues, Asp40, Asp43, Asp93, Ala94 and Asn95, that bind to $_1Ca^{2+}$ and residues, Asp40, Asp43, Asn65 and Thr41, that bind to $_4Ca^{2+}$, as well as a residue Asn65 supports $_3Ca^{2+}$ for stabilizing fold[37]. Moreover, our method predicted a high score (0.6) for the residue Tyr96, which plays a crucial role in lipid headgroup recognition through cation-$\pi$ interaction with the phosphatidylcholine trimethylammonium group[37]. We also applied PhiGnet to $\alpha$LA, which contains a single, tightly bound calcium ion that is cradled in the EF-hand motif to stabilize the protein against denaturation[38]. In the $\alpha$LA protein, the important motif is computationally characterized by a constellation of residues: Lys79, Asp82, Asp84, Asp87, and Asp88. In Ecl18kI, the major groove contacts the bases of the recognition sequence through the three consecutive residues Arg186, Glu187 and Arg188. Specifically, Arg186 and Arg188 form bidentate hydrogen bonds to the outer and inner guanines, respectively. The side chain oxygen atoms of Glu187 each accept one hydrogen bond from the two neighboring cytosines of the recognition sequence. Moreover, the sequence-specific minor groove contacts are exclusively mediated by Glu114[39]. To evaluate the importance of each residue in Ecl18kI, we computed the activation scores for each residue. These scores confirmed the agreement between the residues captured by PhiGnet and those identified through experimental data. For the proteins MCM1-TR and FosJun, our method captured residues with top activation scores that bind to DNAs, although not all of the residues at functional sites were characterized by high probabilities. Meanwhile, the activation scores failed to highlight function-relevant sites for a few residues. For instance, few residues with scores >0.5 were not located at the functional sites in Ribokinase, $\alpha$LA, and HPUK. This discrepancy could be attributed to the noise present in EVCs. Together, the activation scores can indicate essential ligand-/ion-contacting residues, suggesting that learning from diverse levels of evolutionary knowledge can identify binding interfaces at the residue level. Such capability would be valuable in discerning interfaces both inter- and intra-proteins, even in the absence of structural information. Moreover, the predictions suggest that learning from evolutionary knowledge enables us to understand residues arranged in highly ordered patterns, relevant to diverse binding activities. On the other hand, biases originating from the evolutionary data could obscure the activation scores for accessing the functional significance of residues. Collectively, the activation scores can underscore essential ligand-/ion-contacting residues, indicating that learning from diverse levels of evolutionary knowledge can effectively identify binding interfaces at the residue level. Conversely, noise originating from the evolutionary data could influence the activation scores, potentially leading to biases in the identification of functional sites.

## PhiGnet outperforms other state-of-the-art methods
To assess the predictive performance of PhiGnet, we implemented the method to infer function annotations (EC numbers and GO terms) for proteins in the two benchmark test sets (see Methods). We proceeded to compare our method against state-of-the-art methods, including alignment-based methods (BLAST[18], FunFams[40], and Pannzer[41]), deep learning-based methods (DeepGO[25], DeepFRI[21], DeepGOWeb[42],

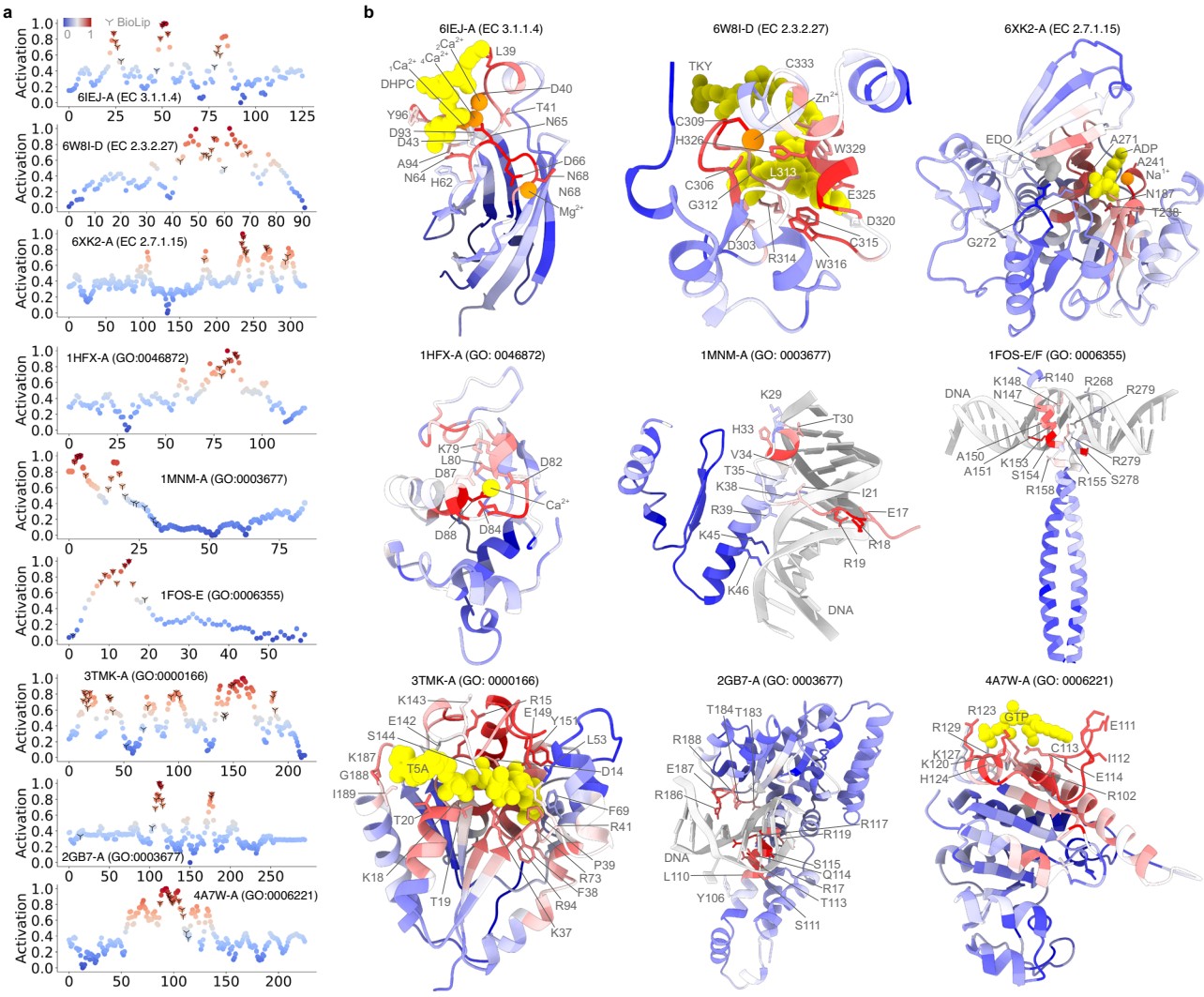

**Fig. 2 | PhiGnet annotates protein function at the residue level. a** The activation score of each residue is predicted using PhiGnet and compared to the biologically relevant ligand-protein binding sites from the BioLiP database. **b** The activation scores are mapped to the tertiary structures of nine proteins, including (left to right, top to bottom) the c2-domain of cytosolic phospholipase A$_2\alpha$ (cPLA$_2\alpha$, PDB ID: 6IEJ)[37], Tyrosine-protein kinase BTK (TpK-BTK, PDB ID: 6W8I), Ribokinase (PDB ID: 6XK2), alpha-lactalbumin (αLA, PDB ID: 1HFX)[38], MCM1 transcriptional regular (MCM1-TR, PDB ID: 1MNM)[60], the Fos-Jun heterodimer (FosJun, PDB ID: 1FOS)[61], the thymidylate kinase (TmpK, PDB ID: 3TMK)[62], Ecl18kI (PDB ID: 2GB7)[39], and helicobacter pylori uridylate kinase (HPUK, PDB ID: 4A7W)[63]. Source data are provided as a Source Data file.

ProteInfer[43], SPROF-GO[44], ATGO+[45], and CLEAN[46]). Two essential metrics, including the protein-centric F$_{max}$-score and the area under the precision-recall curve (AUPR), were utilized for the comparisons. Our method demonstrated predictive capabilities for assigning function annotations to proteins across the two test sets. It achieved an average AUPR of 0.70 and 0.89, as well as F$_{max}$ scores of 0.80 and 0.88, for GO terms and EC numbers, respectively (Fig. 3). Moreover, it consistently maintained strong performance, with average AUPR scores of 0.64, 0.65, and 0.80, alongside corresponding F$_{max}$ values of 0.82, 0.75, and 0.81, for the three branches of GO terms – CC, BP, and MF (Fig. 3d). Overall, PhiGnet significantly outperformed all supervised and unsupervised approaches across the benchmark datasets. For example, in the benchmark of EC numbers, we compared the predictions of various methods, including BLAST, FunFams, DeepGO, DeepFRI, Pannzer, ProteInfer, and CLEAN, against experimentally determined function annotations across the test proteins. Our method yielded F$_{max}$ score of 0.88 and AUPR of 0.89, surpassing the performance of other approaches (Fig. 3a, b, Supplementary Fig. S3). All the compared methods exhibited various performances, as illustrated in the precision-recall curves. DeepFRI, Pannzer, and ProteInfer achieved

a similar F$_{max}$ score, approximately 0.68, outperforming BLAST and DeepGO. In terms of AUPR, FunFams, DeepFRI, and CLEAN yielded similar performances, which were better than those of ProteInfer and Pannzer. PhiGnet achieved F$_{max}$ of 0.88 and AUPR of 0.89, respectively, outperforming the CNN-based DeepGO (F$_{max}$ of 0.37 and AUPR of 0.21), structure-based DeepFRI (F$_{max}$ of 0.69 and AUPR of 0.70), and the contrastive learning-based CLEAN (F$_{max}$ of 0.76 and AUPR of 0.70) (Fig. 3a, b, Supplementary Fig. S3). These results suggest that PhiGnet has the ability to achieve accurate assignment of EC numbers to proteins. In the benchmark of GO terms, we compared our method against nine state-of-the-art methods, utilizing the same metrics to evaluate their performance. Across predictions of CC, BP, MF ontologies, PhiGnet achieved F$_{max}$ of 0.82, 0.75, 0.81 and AUPR of 0.64, 0.65, 0.80, respectively, which are significantly better than those of the compared methods. Notably, although ensemble-networks-based ProteInfer outperformed the remaining approaches over MF and BP ontologies, and the alignment-free SPROF-GO and structure-based DeepFRI excelled over CC ontology, PhiGnet's performance remained superior (Fig. 3d, e, Supplementary Figs. S4–S7, and Table S1). Comparing predictive performances on the GO terms, we found that PhiGnet

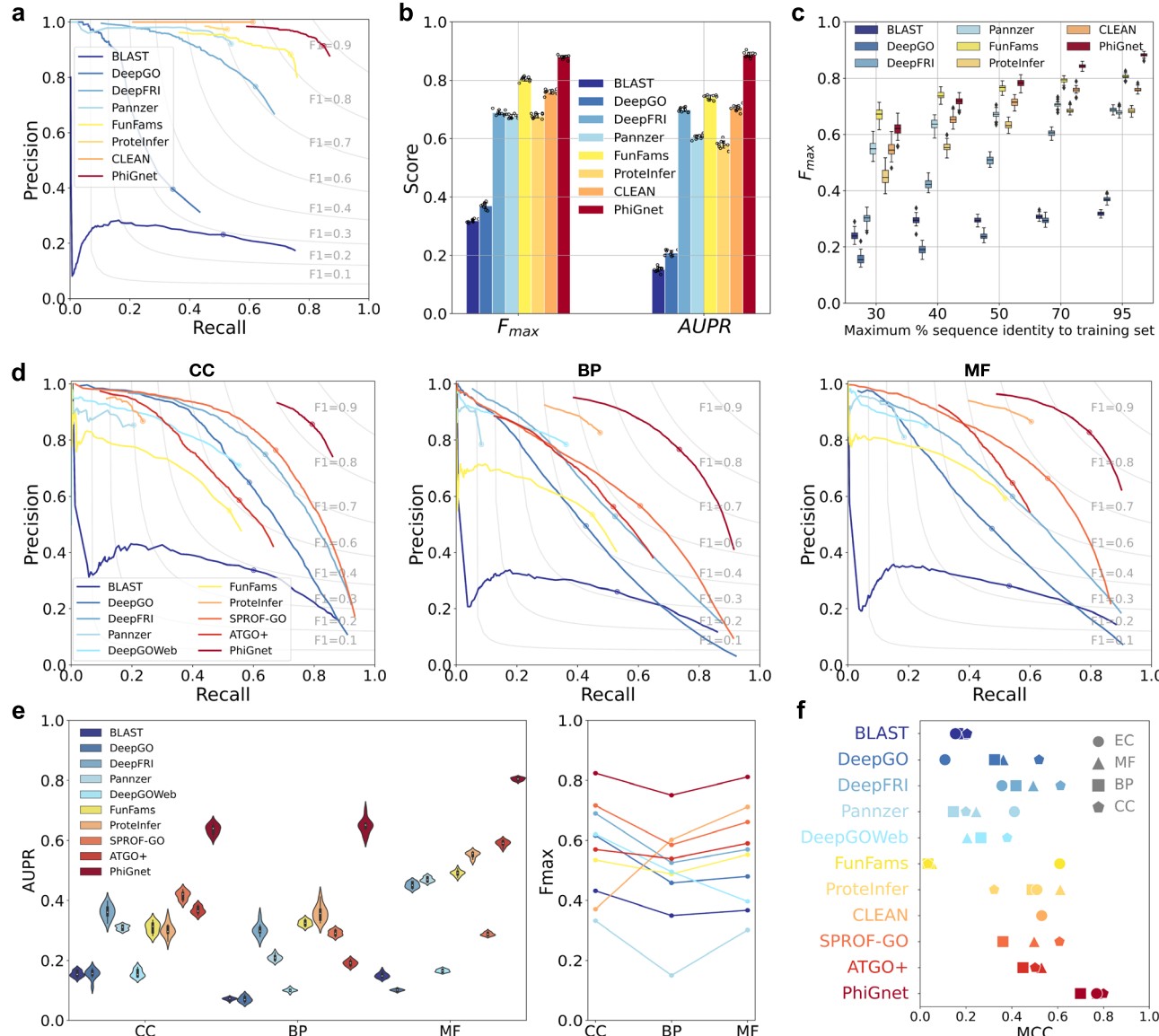

**Fig. 3 | Comparisons among different methods across GO terms in various ontologies and EC numbers. a** Precision-recall curves illustrate the performance of different methods in predicting EC numbers for proteins. **b** Protein-centric $F_{max}$ scores and function-centric AUPR scores are computed across all test proteins to predict EC numbers, where the scores are presented as mean values with standard deviations of 10 bootstrap iterations. **c** Evaluation of robustness in predicting EC numbers as sequence identity increases, where the $F_{max}$ scores of each method at different sequence identities are depicted as boxplots of 50 bootstrap iterations, with the median values at the center and the interquartile range shown by the upper and lower edges of the boxes. **d** Precision-recall performance across GO terms in different ontologies. **e** Left, violin plots showing AUPR with the median values at the center of the distribution of 10 bootstrap iterations, and right, $F_{max}$ scores for the different methods in predicting CC, BP, and MF. **f** Computed Matthews correlation coefficient between predicted scores and ground-truth values for both EC numbers and GO terms using different methods. Source data are provided as a Source Data file.

achieved first place in both accuracy and robustness, significantly better than the eight methods above and another prediction from a web server, DeepGOWeb (Fig. 3d–f).

Moreover, we demonstrated the robustness of PhiGnet for generalization to test proteins with varying thresholds of sequence identity compared to the proteins in the training set. At various maximum sequence identity levels (30%, 40%, 50%, 70%, and 95%), PhiGnet exhibited improved predictive performance as sequence identity increased (Fig. 3c, Supplementary Fig. S5). PhiGnet has been ranked among the top two robust methods for the test set of EC numbers, demonstrating consistently predictive performance with $F_{max}$ values of 0.61 and 0.72 at sequence identity levels of 30% and 40%, respectively. When compared to the domain-based method FunFams ($F_{max}$ of 0.67 and 0.74), PhiGnet slightly underperformed at sequence identity

thresholds of 30% and 40%. However, PhiGnet achieved comparable or better performance when sequence identity exceeded 50%. Similarly, the performance of DeepFRI, FunFams, ProteInfer, and CLEAN also improved as sequence identity increased. Pannzer exhibited a similar trend when sequence identity was below 50%, but its performance remained nearly constant with a slight decrease in $F_{max}$. In contrast, both BLAST and DeepGO showed slight improvements as the proteins in the test set increased sequence identity to those in the training set. The robustly predictive performance of PhiGnet has also been demonstrated by predicting the three branches of GO terms, maintaining high accuracy even at low sequence identity (Supplementary Fig. S5). In predictions of both EC numbers and GO terms, we also calculated the Matthew's correlation coefficient (MCC) between the predicted scores and ground truth to quantitatively compare the

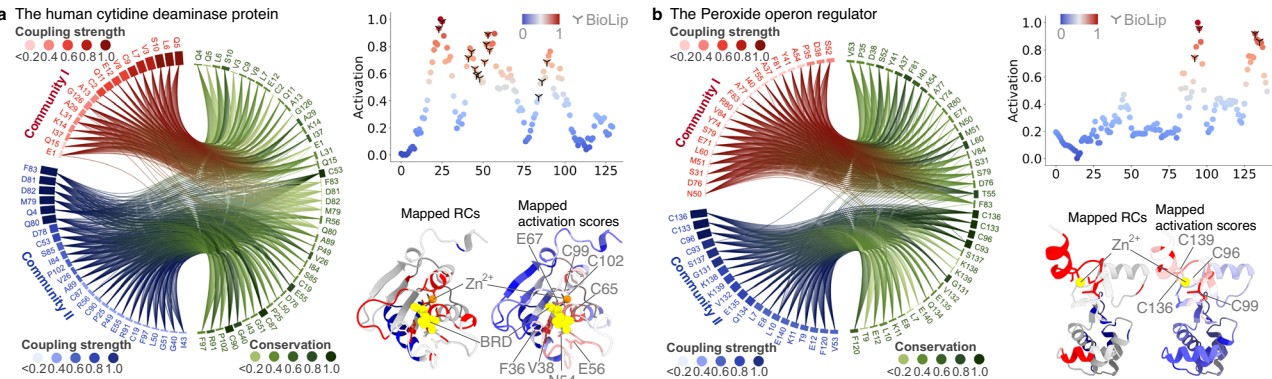

**Fig. 4 | PhiGnet learns evolutionary signatures for identification of protein functional sites.** Mappings of RCs and activation scores of (**a**) the human cytidine deaminase protein (hCDA, PDB ID: 1MQ0-A, GO term 0008270), and (**b**) the Peroxide operon regulator (PDB ID: 2FE3-A, GO term 0046872). The residues within each RC are shown in the chord plotting with coupling strength and degree of conservation in bars. The activation scores (dotted lines) of each protein are compared to the BioLip identifications (marked with Y in black), and residues with high scores (in red) are also compared to those within RCs on their 3D structures. The 1-beta-ribofuranosyl-1,3-diazepinone (BRD) and $Zn^{2+}$ ions are shown with spheres in yellow (orange for the $Zn^{2+}$ ion in hCDA). Source data are provided as a Source Data file.

performance of various methods. PhiGnet achieved an average MCC of 0.76, which is higher than the average MCCs of the other ten state-of-the-art methods (Fig. 3f).

## PhiGnet driven by evolutionary signatures

The evolutionary data plays an important role in PhiGnet for predicting protein function annotations and identifying functional sites. First, we performed ablation experiments to test how EVCs/RCs contribute to PhiGnet. We trained PhiGnet using either EVCs or RCs alone and assessed its performance in terms of $F_{max}$-score and AUPR over predictions of EC numbers/GO terms. To accomplish this, we chose a threshold (0.2) for both EVCs and RCs based on the similar performances in predicting EC numbers/GO terms (Supplementary Fig. S8), aiming to mitigate potential noise arising from coevolution or weak couplings between pairwise residues. We first test whether the information in EVCs, which preserve evolutionary couplings at sites of co-variation, is sufficient to infer functional annotations. The second experiment tests the necessity of information in RCs that independently capture high-order couplings. Similarly, we built a model using RCs alone to computationally assign functional labels to proteins, and this model produced slightly better predictions (Supplementary Figs. S9 and S10). The two experiments indicate that both models demonstrate the capability to accurately assign functional annotations to proteins. Moreover, PhiGnet, utilizing either EVCs or RCs, demonstrates a robust capacity to learn general sequence-function relationships, often better than or as good as other approaches, even test proteins exhibiting low sequence identity in presence of the training set (Fig. 3c, Supplementary Figs. S9c and S10c). Through precision and robustness comparisons, we have demonstrated that the evolutionary signatures (EVCs and RCs) constitute crucial attributes capable of enhancing deep learning-based methods for protein function annotations.

Secondly, we asked whether the residues, particularly within RCs that are often relevant to the specific function, can be quantified for functional sites. To address this, we further investigated the capability of PhiGnet to characterize meaningful features from the identified function-relevant residues within the residue communities. The activation scores were computed for the residues to underscore their contributions to the protein function. Notably, the predicted residues concurred with those at the functional sites identified through experimental determinations, better identifications than those in RCs (Fig. 4). In the human cytidine deaminase (hCDA) protein[47], compared to residues within RCs that were identified as functionally relevant,

PhiGnet quantitatively characterized their importance in the binding between hCDA and $Zn^{2+}$/BRD through more accurate predictions of active sites: Cys65, Cys99, and Cys102, which coordinate with the zinc ion, as indicated by the activation scores (Fig. 4a). In the Peroxide operon regulator (PerR), we also observed that PhiGnet narrowed down the number of residues located within RCs[48] and effectively distinguished non-$Zn^{2+}$-binding residues from the binding ones, compared to RCs. Specifically, Cys96, Cys99 and Cys136, Cys139 exhibited much higher activation scores. These residues collectively coordinate the zinc ion, locking the three $\beta$-strands together to form the arrangement of the dimeric $\beta$-sheet, in contrast to the non-binding residues (Fig. 4b). In light of these results, we conclude that the evolutionary information, particularly that contained in RCs, is sufficient to specify a protein's function and to quantitatively characterize the residues at the functional sites. Moreover, the results argue that RCs contain evolutionary knowledge at a higher-ordered level than the information in EVCs at a lower-ordered level. Meanwhile, information contained in RCs plays an important role in enhancing PhiGnet's ability to identify functionally relevant sites at the residue level.

## Test on CAFA3 targets

To assess whether the different performances of the methods under evaluation, and the superiority of PhiGnet were inherent to the algorithms or due to different training sets, we re-executed two alignment-based methods (BLAST and FunFams) and conducted retraining on four deep learning-based methods (DeepGO, ATGO+, SPROF-GO, and PhiGnet). Other methods were excluded primarily due to the unavailability of trainable source codes or because such method required unavailable structural information) against an identical dataset. We used the third Critical Assessment of Protein Function Annotation (CAFA3) dataset consisting of 66,841 proteins[49]. To address homology issues, proteins sharing over 30% sequence identity with the test proteins were excluded from the training dataset[45]. The remaining proteins were utilized to construct databases for BLAST and FunFams. 95% of them were randomly selected for training DeepGO, ATGO+, SPROF-GO, and PhiGnet, with the remaining 5% reserved for validation to fine-tune the methods' parameters. Moreover, we conducted comparisons among the different methods using the CAFA3 test proteins either with less than 60% sequence identity to those in the training dataset or without redundancy removal (Supplementary Fig. S12).

A comparison among the six different methods implemented on the CAFA3 dataset reveals that PhiGnet exhibits the best performance across both $F_{max}$ and AUPR metrics (Table 1, Supplementary Fig. S12).

**Table 1 | Comparison of different methods on the CAFA3 dataset**

| Method | $F_{max}$ | | | AUPR | | |
|---|---|---|---|---|---|---|
| | BP | CC | MF | BP | CC | MF |
| BLAST | 0.264 | 0.390 | 0.394 | 0.087 | 0.205 | 0.228 |
| FunFams | 0.372 | 0.272 | 0.440 | 0.162 | 0.115 | 0.316 |
| DeepGO | 0.313 | 0.467 | 0.230 | 0.165 | 0.437 | 0.180 |
| ATGO+ | 0.458 | 0.493 | 0.470 | 0.378 | 0.361 | 0.323 |
| SPROF-GO | 0.480 | 0.358 | 0.522 | 0.397 | 0.225 | 0.421 |
| PhiGnet | 0.531 | 0.584 | 0.606 | 0.425 | 0.590 | 0.571 |

PhiGnet achieved the highest $F_{max}$ scores across all three categories: BP (0.531), CC (0.584), and MF (0.606), indicating its superior capability in predicting functional annotations across diverse biological processes, cellular components, and molecular functions compared to methods such as BLAST, DeepGO, FunFams, and ATGO+. Furthermore, PhiGnet outperformed other methods with AUPR scores of 0.425 for BP, 0.590 for CC, and 0.571 for MF, demonstrating its effectiveness in accurately identifying true positive annotations while minimizing false positives across various functional categories. Although methods like BLAST, DeepGO, FunFams, and ATGO+ exhibited respectable performance in specific categories, none consistently achieved high scores across both $F_{max}$ and AUPR metrics as PhiGnet did. Overall, the comparison underscores PhiGnet as one of the state-of-the-art methods on the CAFA3 dataset, demonstrating that its increased performance is independent of the training dataset used.

## Predicting functions of holdout and unannotated proteins

Can PhiGnet annotate uncharacterized proteins? We carried out our predictions for the independent hold-out set of 6229 proteins (Supplementary Fig. S13). We followed the same procedures to collect EVCs, RCs, and sequence embeddings for all the proteins. They were utilized to feed into the fine-tuned PhiGnet in order to compute a probability tensor for assigning functional annotations to the proteins. Among the collected proteins, our method's overall performance was superior to that of state-of-the-art methods. Given that these proteins were independently collected, our computational predictions can be valuable in assigning functional annotations to new proteins (Supplementary Figs. S14, S15, and Table S2). For example, across the T. forsythia NanH (PDB ID: 7QXO) and human Sar1b (PDB ID: 8E0A), the activation scores successfully indicate the functional sites that bind to Oseltamivir and guanosine tetraphosphate (Supplementary Fig. S16). Our analysis shows that PhiGnet's high confidence prediction is in a good agreement with experimental annotations, suggesting that it would contribute to computational efforts for assigning function annotations to proteins with unknown labels. This applies even when dealing with experimental annotations of lower confidence scores, and can benefit experimental investigations of different biological activities. Moreover, by leveraging evolutionary information, PhiGnet provides function annotations as well as residue-level activation scores for over 2.5 million individual sequences within the UniProt database. The activation score assigned to each individual residue offers a quantitative measure of its significance in a specific activity, proving beneficial for screening experiments aimed at identifying functionally important sites.

## Discussion

It has been long appreciated that investigating evolutionary information across species can further our understanding of protein function and of the consequences of pathological mutations, even at the residue level. By leveraging deep learning methods on continuously expanding sequencing data, we can extract valuable knowledge to accurately annotate protein functions. This can greatly benefit both biological and clinical research, as well as facilitate drug discovery.

We have demonstrated that a statistics-informed learning method trained solely on evolutionary data achieves state-of-the-art performance in predicting protein function annotations at the residue level. The approach presented here requires no inputs other than the protein sequence and learns its characterized embedding using the statistics-informed graph convolutional networks. We show that EVCs and RCs have crucial effects on the predictions of protein function annotations and on the identifications of residues at functionally relevant sites. Our method produces high-accuracy annotations and identifies functional sites at the residue level. Therefore, this approach is well-suited for gaining a better understanding of the biological activities of unannotated or poorly studied proteins, as well as for quantitatively investigating the effects of disease-related variants.

When evaluating the performance of the methods presented (see Fig. 3), it becomes evident that PhiGnet outperforms its counterparts due to its distinctive amalgamation of two key factors. Firstly, it integrates insights derived from both evolutionary coupling analysis and spectrum analysis, resulting in a more comprehensive grasp of the intricate relationship between protein sequences and their functions. In contrast, other methods, such as FunFams and Pannzer, predominantly rely on homology-based approaches. Although homology-based methods have their merits, they might not capture the subtle nuances and intricate connections between proteins that are unveiled by the evolutionary coupling data. Conversely, while DeepFRI, DeepGO, SPROF-GO, and ATGO+ depend on structural data and homologous information, they may not harness the same depth of evolutionary data as PhiGnet. Moreover, the spectrum analysis applied to evolutionary data delves into the high-order patterns within protein sequences, which also contributes to PhiGnet's superior performance. Secondly, although DeepFRI, DeepGO, SPROF-GO, ATGO+, and CLEAN are effective in leveraging pre-trained models for protein function prediction, PhiGnet distinguishes itself by enhancing the pre-trained model with evolutionary insights. This augmentation enables PhiGnet to offer a more holistic perspective on protein functions. By combining the ESM-1b model with evolutionary knowledge, PhiGnet achieves a deeper and more comprehensive understanding of the intricate relationship between protein sequences and their functions. This unique combination gives PhiGnet a competitive edge in accurately assigning EC numbers or GO terms to proteins, as it taps into a broader array of evolutionary features that many other methods do not fully explore.

In conclusion, the better performance of PhiGnet can be attributed to its utilization of the evolutionary data and high-order patterns of the data from protein sequences, allowing for a deeper and more accurate understanding of protein functions. PhiGnet leverages physically-inferred knowledge (EVCs and RCs) and performs significantly better predictions across both benchmark test sets of EC numbers and GO terms. This underscores PhiGnet's capacity to effectively assimilate enriched evolutionary knowledge, where protein function has evolved and been encoded, to delineate the intricate relationship between protein sequences and their functions. Moreover, PhiGnet achieved higher accuracy in $F_{max}$ compared to the other approaches, even when dealing with proteins in the test set with low sequence identity to those in the training set. These comparisons lead us to conclude that PhiGnet demonstrates the capability for generalization in predicting protein function annotations across both EC numbers and GO terms.

The primary success of our approach lies in the utilization of statistics-informed graph convolutional neural networks to facilitate hierarchical learning over evolutionary data from massive sequence datasets. This approach surpasses existing supervised and unsupervised methods significantly and may be used to guide future biological and clinical experiments. We are aware that machine learning-based methods are highly dependent on the datasets that are used to

tune their parameters. To mitigate bias arising from the datasets, it is important to curate proteins for training, maintain diversity in sequences, and evaluate the methods on various proteins to assess their generalization capabilities. Limitations of our method might include biases/noise arising in protein families with less diverse sequences. Incorporating (co-)evolutionary information into PhiGnet can impact the accurate identification of residue communities, particularly if the information is derived from a highly conserved protein family. While integrating physically extracted knowledge into our method yields a significant improvement compared to other approaches, there are still significant challenges in interpreting the learning mechanisms within PhiGnet. For instance, a protein might have more than one active or functionally relevant sites. The activation score does not allow to discern active site a given residue is part of.

We anticipate that evolutionary information will enable statistics-informed learning approaches to effectively characterize protein function at the residue level, including predicting disease variants, allosteric regulation, binding affinity, and specificity from sequence alone, as well as incorporating structural information for specific applications. The synergy between evolutionary data and machine learning will pave the way for accurately determining and engineering the biophysical properties of proteins, with implications spanning clinical decisions, industrial applications, and environmental biotechnology.

# Methods

## Datasets
In the present study, we collected protein chains from the Protein Data Bank (PDB)[50] using the protocols[21] to construct datasets (until 10/2021). The collected protein chains were clustered at 95% sequence identity. From each cluster, we selected a representative protein possessing at least one annotated function. Two benchmark datasets were created, comprising 41,896 and 20,215 protein chains (with a maximum of 1024 residues each), annotated with GO terms and EC numbers, respectively. In the benchmark of EC numbers, we extracted unique annotations from the third-/fourth-level of the proteins, forming a total of six primary catalytic reaction classes: oxidoreductase, transferase, hydrolase, lyase, isomerase, and ligase. For the benchmark of GO terms, the three categories, BP, CC, and MF, are utilized to evaluate and compare the performance of various methods in this study. In the present study, we divided each dataset into three subsets, including training, validation, and test sets, with ratios of 8:1:1, respectively. The protein sequences in the test set (Supplementary Fig. S17) are of varying degrees (30%, 40%, 50%, 70%, and 95%) of sequence identity against that in the training set.

To create an independent hold-out set, we collected 13,584 proteins that are released after 1/2022 from the RCSB PDB database[50] (released between 1/2022 and 12/2022). Subsequently, we then searched these proteins against the SIFT database[51] (as of December 2022) to filter out proteins lacking experimentally determined functional annotations. As a result, we obtained 6229 proteins of less than 1024 residues as an independent hold-out test set. We implemented the trained PhiGnet to assign function annotations to these recently released proteins, and the predictions are evaluated against the annotations in the SIFT database.

## Characterizing evolutionary signatures
To calculate evolutionary couplings, we collected an MSA for the target protein by searching its sequence against the UniClust30 database (up to February 2022)[52] using the *hhblits* tool[53] (version 3.3.0) with default parameters. Afterward, we performed trimming on each MSA using in-house scripts to eliminate sequences of low quality (for instance, sequences with over 80% gaps were removed). The distributions of MSA quality were obtained for both the training and test sets (Supplementary Fig. S18). For each of the trimmed MSAs, we

utilized our in-house scripts based on *leri*[12] to compute EVCs between pairwise residues. Subsequently, we derived RCs that capture functional signatures from these couplings. Both evolutionary couplings and residue communities were used as graph edges within PhiGnet in predicting protein functions. The computed EVCs may contain noise arising from the coevolution of residues across different sequences[54]. As a result, we implemented a normalization process on all computed EVCs, using a threshold of 0.2 to enhance their quality. Likewise, the scores within the RCs were also normalized to fall within the [0, 1] range and were subjected to filtering using a threshold of 0.2. These actions were informed by the experimental design's focus on hyper-parameter optimization through grid search (Supplementary Fig. S8).

## Learning information using the ESM-1b transformer
To allow evolutionary diversity of natural sequences, we leveraged the pre-trained model ESM-1b transformer[31] as physically embedded knowledge (across 250 million protein sequences) to improve the prediction ability of PhiGnet. The ESM-1b transformer is pre-trained on UniRef50 representative sequences and a specialized embedding of protein sequences to represent biological information at multiple levels, e.g., evolutionary homology. In this study, we derived the embedding of the provided protein sequence from the ESM-1b transformer's output. This embedding was then integrated with EVCs and RCs to feed into PhiGnet. The integrated strategy offers insights into remote protein homology, leveraging informative relationships within the embedding representations of homologous proteins. This allows for generalization to previously unseen proteins in the training set.

We encoded each protein sequence using a sequence-level embedding from the ESM-1b model. Each amino acid is represented by a one-hot feature vector and embedded as an input representation for PhiGnet. The ESM-1b embedding captures the unique amino acid at each specific site along the sequence, enabling the stacked GCN layers to acquire higher-level features from either EVCs or RCs using distinct convolutional filters.

## Statistics-informed graph networks
PhiGnet adopts dual channels consisting of stacked GCNs. In one channel, a stack of GCNs gathers information from the sequence embedding using evolutionarily coupled residues as graph nodes. In the other channel, the graph layers learn information about functionally significant residues using RCs as nodes. The PhiGnet architecture is composed of six GCN layers and two fully connected layers with dropout. Initially, a protein sequence of interest is used to compute EVCs, RCs, and the ESM-1b embedding information[31]. The first layer of each channel loads tensors of $L \times 1,280$ from sequence embedding, and a tensor of EVCs/RCs is used as the adjacency matrix throughout all the three stacked graph layers (Fig. 1a). In the two channels, EVCs are to describe the linkage between pairwise residues, while RCs are used to characterize hierarchical interactions for the other three stacked graph layers (Supplementary Fig. S19). They motivate PhiGnet to learn knowledge of residues that significantly contribute to protein function. The final fully connected layer incorporates a fixed-number SoftMax layer to compute the prediction probability for assigning function annotations to the protein.

In PhiGnet, we embed the given sequence of $L$ amino acids using the ESM-1b transformer as a tensor $\mathbf{T}_{esm}$ ($\mathbf{T}_{esm} \in \mathbf{R}^{L \times D}$, $D$ is the dimension of the tensor). The sequence embedding is the input of the two channels in GCN to represent graphs at different levels, and we employ two adjacency matrices (EVCs and RCs) to describe the linkages between residues at two different levels. In each GCN layer of PhiGnet, we employed an undirected connected graph $\mathbf{G} = \{\mathbf{V}, \mathbf{E}, \mathbf{A}\}$, consisting of a set of nodes $\mathbf{V}$ with $L$ residues, a set of edges $\mathbf{E}$ defined by the adjacency matrix $\mathbf{A}$ (a matrix of EVCs or RCs is used in the present study). If residue $i$ is correlated with residue $j$ as defined by the entry $\mathbf{A}(i, j) = 1$; otherwise, there is no edge between residues $i$ and $j$,

$\mathbf{A}(i,j) = 0$. The degree of the matrix $\mathbf{A}$ is denoted as a diagonal matrix $\mathbf{D}$, where $\mathbf{D}(i,i) = \sum_{j=1}^{n} \mathbf{A}(i,j)$. Each GCN layer involves two phases of aggregation, where each node gathers and aggregates features of its neighbor nodes to update the local features, and combination, where the updated features are further merged to extract high-level abstraction through a local multilayer perceptron network. The layer-wise forward propagation of GCN is defined as follows,

$$f\left(\mathbf{H}^{(k+1)}, \mathbf{A}\right) = \sigma\left(\mathbf{A}\mathbf{H}^{(k)}\mathbf{W}^{(k)}\right), \tag{1}$$

where $\mathbf{H}^{(k)}$ and $\mathbf{W}^{(k)}$ are the representation of residues and weights of the $k$th layer, respectively, and $\sigma(\cdot)$ non-linear activation functions. In the present study, we implemented a normalized form over GCN and essentially arrive at the propagation rule[55]:

$$f\left(\mathbf{H}^{(k+1)}, \mathbf{A}\right) = \sigma\left(\hat{\mathbf{D}}^{-\frac{1}{2}}\hat{\mathbf{A}}\hat{\mathbf{D}}^{-\frac{1}{2}}\mathbf{H}^{(k)}\mathbf{W}^{(k)}\right), \tag{2}$$

with $\hat{\mathbf{A}} = \mathbf{A} + \mathbf{I}$, where $\mathbf{I}$ is an identity matrix and $\hat{\mathbf{D}}$ is the diagonal node degree matrix of $\hat{\mathbf{A}}$.

There are three blocks of GCN layer that are used in each channel of PhiGnet, and the number of hidden units in each GCN layer is set to 512. Information extracted by different channels, using either EVCs or RCs, can promote PhiGnet to learn features at two levels (Supplementary Figs. S9–S11). The outputs of the GCNs are concatenated into a tensor of dimensions $L \times D$, where $L$ represents the number of nodes in the graphs. To consolidate the information across the $L$ dimension, we apply a SumPooling layer, reducing $L$ to 1 while preserving the other dimension. This aggregated tensor of size $1 \times D$ is forwarded to the FC layers for predicting protein functions.

## Hyper-parameter tuning and PhiGnet training

The present PhiGnet allows us to directly learn information from a sequence alone (without using any structural knowledge) to significantly explore functional sites at the residue level. To achieve an optimized model, we have to tune and choose values of the hyper-parameters in our method, e.g., thresholds for filtering EVCs/RCs (Supplementary Fig. S8). This tuning of parameters is crucial to guarantee both the stability and performance of PhiGnet.

With the pre-defined hyper-parameters, we implemented a cross-entropy loss function to balance the abilities of learning and generalization. The loss function is defined as follows,

$$\mathcal{L} = -\frac{1}{N}\sum_{i=1}^{N}\sum_{j=1}^{F}\left[y_{ij}\log(\hat{y}_{ij}) + (1-y_{ij})\log(1-\hat{y}_{ij})\right], \tag{3}$$

where $N$ is the number of data samples, and $F$ is the number of function classes in EC numbers/GO terms. $y_{ij}$ is to label the ground truth to 1 if the $i$th sample is in the $j$th function class, otherwise, it is 0. Similarly, $\hat{y}_{ij}$ is a label for the prediction.

PhiGnet was trained with batch size of 64 for maximum 500 epochs using early-stopping criterion over the defined cross-entropy loss (Eq. (3)). During training, we leveraged the Adam optimizer[56] with a learning rate of $2 \times 10^{-4}$, $\beta_1 = 0.9$, $\beta_2 = 0.999$, $\epsilon = 1 \times 10^{-6}$, and $L_2$ weight decay of $2 \times 10^{-5}$. To avoid over-fitting, we implemented a dropout of 0.3 for the second fully connected layer. Accordingly, we achieved fine-trained models of PhiGnet that are leveraged to predict the probability of assigning EC numbers/GO terms to a given protein by learning from sequence embedding under constraints of evolutionary couplings and couplings intra residue communities.

## Function annotations at the residue level

To quantitatively evaluate the importance of residues, we implemented the gradient-weighted class activation map method (that localizes the most important regions in images relevant for making correct classification decisions in computer vision)[32] for a specific function annotation to compute scores that are assigned to each residue in a protein. In the grad-CAM method, the gradient information of a given layer is used to compute localization map $\mathbf{M}^c \in \mathbb{R}^{u \times v}$ with width $u$ and height $v$, and it is used to characterize the importance of every single element of the input for a specific class $c$. Given a feature map $\mathbf{F}^k$, the activation value $\mathcal{S}^c$ for scoring the class $c$ is computed to measure the importance of neurons, $\alpha_k^c$, as follows,

$$\mathcal{S}^c = \text{ReLU}\left(\sum_k \alpha_k^c \mathbf{F}^k\right), \tag{4}$$

$$\alpha_k^c = \frac{1}{L}\sum_i^L \frac{\partial Y^c}{\partial \mathbf{F}_i^k}, \tag{5}$$

where $\text{ReLU}(\cdot)$ is a non-linear activation function, holding a positive effect for function class $c$, and $L$ is the number of elements in the input.

In the present method, we evaluated the importance of the $i$th amino acid in the feature map $\mathbf{F}^k$ obtained from the layer concatenated from the two channels in PhiGnet, and the gradient $\frac{\partial Y^c}{\partial \mathbf{F}_i^k}$ is calculated by the derivative of the function annotation $c$ with predicted score $\mathbf{Y}^c$, with respect to the feature map $\mathbf{F}_i^k$ in sequence of length $L$.

## Comparison with existing approaches

In the present study, we compared our method to eight methods, including BLAST[18], FunFams[40], DeepGO[25], DeepFRI[21], ProteInfer[43] ATGO[45], SPROF-GO[44], and CLEAN[46] in details. Moreover, our method was compared to predictions collected from two web-servers, DeepGOWeb[42] and Pannzer[41], over predictions of either GO terms in different ontologies or EC numbers using the collected data sets.

BLAST is a sequence searching tool based on the local sequence alignment algorithm[18]. Implementing BLAST, we transferred function annotations to proteins within the test set from all the annotated sequences in the training dataset following the same procedure as presented in refs. 20,21. The probability assigning annotation(s) to each protein was computed by sequence identity in percentage between the sequences in the test and training sets. More specifically, if a protein in the test set hits against proteins in the training set with a maximum sequence identity of 75%, it was assigned function annotation(s) by transferring all the annotations from training proteins with a score of 0.75. In practice, we filtered out sequences from the training set using default parameters to keep within limits of assigning annotation(s) from homologous sequences[21].

FunFams is a domain-based approach that leverages CATH superfamilies to transfer function annotation from a protein to another[40]. Given a protein, its sequence is searched against the CATH using the HMMER tool[57], and its function annotation (EC numbers and GO terms) is copied from the FunFams with the highest HMM score. We obtained EC numbers and GO terms for the test proteins in this study by following the procedure present at https://github.com/UCLOrengoGroup/cath-tools-genomescan. More specifically, each protein is assigned a score (measuring either GO terms or EC numbers) that is computed from the frequency of proteins from the sequence alignment collected by FunFams from the CATH database.

DeepGO is a supervised deep learning method using convolutional neural networks (CNN) to predict GO terms initially[25]. DeepGO learns features from both protein sequences and a cross-species protein-protein interaction network using a CNN layer with 32 filters. In DeepGO, each protein sequence is encoded as a one-hot embedding and fed into the CNN model to compute sequence representation, combined with the embedding of protein-protein network. With a fully-connected layer of a sigmoid activation function, DeepGO generates a probability as confidence to assign a function annotation the query sequence. For fair comparison, we locally adopted DeepGO with

default settings to predict both EC numbers and GO terms for the test set of proteins.

DeepFRI was constructed based on an architecture of graph convolutional networks to learn both protein sequence using a pre-trained LSTM model and its structural information[21]. DeepFRI leverages the pre-trained LSTM model to extract the feature of sequence, and such feature is learned by the graph convolutional networks using residue contacts that are derived from protein tertiary structure as representations for connections of residues, e.g., the $i$th and $j$th residues are contacted if the distance between the two $C_\alpha$ atoms of the residues is less than a threshold of 10 Å; otherwise, they are not contacted. We locally implemented DeepFRI with its default configurations and collected the protein structures for the test set from the RCSB PDB database[50]. The residue contacts within each protein were computed under the threshold from its structure and used as structural information for DeepFRI to predict EC numbers/GO terms.

DeepGOWeb is developed based on DeepGOPlus[58], an extended variant of the DeepGO method, and it utilizes many convolutional filters of different kernel sizes to learn protein sequence representations. As an improved method, it further embeds homology-based predictions from DIAMOND[59] to improve predictive accuracy. We collected the DeepGOWeb predictions over our test set of proteins from its webs-server with default parameters. We submitted our test protein sequences to the DeepGOWeb web-server and collected the predictions over the test sequences to compute both protein-centric $F_{max}$ score and term-centric AUPR for comparison.

Pannzer is a weighted K-nearest neighbor predictor for assigning function annotations to proteins[41]. Pannzer searches a query sequence against the UniProt database to collect the sequence neighborhood, and the annotations are transferred to the query protein from its homologous neighbors. We collect the Pannzer predictions of EC numbers and GO terms on our test set using its web-server.

ProteInfer is a method based on a single convolutional neural network scan for all known domains in parallel[43]. Proteinfer has 1100 filters to learn the mapping between protein sequences and functional annotations. The method was trained on the well-curated portion of Swiss-Prot data. The finely-tuned ProteInfer maps an amino acid sequence through five residual convolutional layers to create embeddings. These embeddings are then extracted using a fully connected layer featuring an element-wise sigmoid activation function, which facilitates the prediction of per-label probabilities.

SPROF-GO is a sequence-based alignment-free protein function predictor that embeds protein sequences using a pre-trained protein language model[44]. The sequence embedding is acquired through two parallel multi-layer perceptron networks, each designed for different latent representations. Additionally, another multi-layer perceptron is to map these representations to protein function label(s) (GO terms). The final predicted annotations are derived from the network model's predictions and homology information with the training dataset, established using DIAMOND[59].

ATGO adopts a triplet neural-network architecture using embeddings from the pre-trained ESM-1b model[31] to predict protein annotations (GO terms)[45]. In ATGO, the embeddings are generated from the last three layers and fused by a fully connected neural network. The triplet neural-network maps the fused representation to predict the confidence scores of protein GO terms. The ATGO+ method is a combination of the ATGO method and a sequence homology-based method, resulting in superior performance compared to ATGO.

CLEAN has been developed based on the contrastive learning for predictive assignments of EC numbers to enzymes[46]. The CLEAN method learns embedded representations of enzymes, in which proteins of the same EC numbers are close to each in Euclidean distances; otherwise, they are far from each other. The positive and negative samples are defined by the distances to the anchor sequence. Positive samples are closer to the anchor sequence, while negative samples are farther away from the anchor sequence. All sequences are embedded using the pre-trained ESM-1b model[31] and are then fed into a supervised contrastive learning neural network. Both the maximum separation and $P$ value methods are employed to prioritize confident predictions of EC numbers in the final inferred results.

**Performance metrics.** We evaluate the different methods using two metrics: protein-centric maximum F-score ($F_{max}$) that measures the precision of labeling EC numbers/GO terms to a protein and term-centric area under precision-recall (AUPR) curve that measures the precision of labeling proteins to different EC numbers/GO terms. The F-score is the harmonic mean of the precision $p(t)$ and recall $r(t)$, while $F_{max}$ represents the maximum F-score achieved. $F_{max}$ and AUPR were defined as follows,

$$F_{max} = \max_t \left\{ \frac{2 \cdot p(t) \cdot r(t)}{p(t) + r(t)} \right\}, \tag{6}$$

$$AUPR = \int_0^1 p(t) \times r(t)\,dt, \tag{7}$$

where $p$ and $r$ are precision that measures the predictive accuracy and recall that is to measure successfully retrieved information, respectively.

### Statistics and reproducibility
No statistical method was used to predetermine sample size.

### Reporting summary
Further information on research design is available in the Nature Portfolio Reporting Summary linked to this article.

## Data availability
All relevant data supporting the key findings of this study are available within the article and its Supplementary Information files. All crystal structures of proteins used in this study are available at Protein Data Bank (https://www.rcsb.org) under accession codes: 4JDZ [https://doi.org/10.2210/pdb4JDZ/pdb], 6IZW [https://doi.org/10.2210/pdb6IZW/pdb], 6IEJ [https://doi.org/10.2210/pdb6IEJ/pdb], 6W8I [https://doi.org/10.2210/pdb6W8I/pdb], 6XK2 [https://doi.org/10.2210/pdb6XK2/pdb], 1HFX [https://doi.org/10.2210/pdb1HFX/pdb], 1MNM [https://doi.org/10.2210/pdb1MNM/pdb], 1FOS [https://doi.org/10.2210/pdb1FOS/pdb], 3TMK [https://doi.org/10.2210/pdb3TMK/pdb], 2GB7 [https://doi.org/10.2210/pdb2GB7/pdb], 4A7W [https://doi.org/10.2210/pdb4A7W/pdb], 1MQ0 [https://doi.org/10.2210/pdb1MQ0/pdb], 2FE3 [https://doi.org/10.2210/pdb2FE3/pdb], 7QXO [https://doi.org/10.2210/pdb7QXO/pdb], and 8E0A [https://doi.org/10.2210/pdb8E0A/pdb]. The data is available for download at https://doi.org/10.5281/zenodo.12496869. Source data are provided with this paper.

## Code availability
The PhiGnet Python code and pre-trained model are available at: https://doi.org/10.5281/zenodo.12496869.

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

## Acknowledgements

This work was funded by Wellcome Trust (214291/Z/18/Z, to B.K.). We thank members of the Kornmann laboratory and AmoAi for many valuable discussions. Y.J.J. and Q.Q.Q. are supported by AmoAi.

## Author contributions

Y.J.J. led the research, conceived the end-to-end approach, designed experiments, financed the experiments, and wrote the manuscript. Q.Q.Q. collected the data, implemented the method, contributed with principal analysis and wrote the first draft. S.Y.H. conducted principal analysis over predictions. X.M.D. conducted data analysis. A.T.J.P. supported with principal analysis and wrote the manuscript. B.K. led the research, funding acquisition, contributed technical advice, and wrote the manuscript. All authors read the final manuscript.

## Competing interests

Y.J.J. is a founder of AmoAi Technologies, UK. The remaining authors declare that the research was conducted in the absence of any commercial or financial relationships that could be construed as a potential conflict of interests.
