## [Peer Review File · Nature Communications]

Reviewers' Comments:

Reviewer #1:

Remarks to the Author:

This paper introduces an physics (evolutionary) information guided deep learning method for protein function prediction. There are two key novel points of this manuscript, the first one is the leverage of evolutionary couplings and residues communities, and the other is the function prediction at the residue level. Besides, this deep method does not require structure data for function prediction, and it outperforms other representative methods by a large margin with better flexibility. The ablation study also proves the effectiveness and rationality of the usage of evolutionary couplings and residues communities. Overall, this paper is interesting and gives some new insights for protein function prediction. Here are some comments for the authors:

1. It is well known that pre-trained model embody refined functional information for protein function prediction, the proposed PhiGnet uses the ESM-1b to extract the raw sequence data, and feeds it into the prediction model. It is unclear whether other comparing methods also use the ESM-1b as the input of sequence data? If not so, it is unclear whether the improved performance is really from the two evolutionary features or not.
2. As to results of PhiGnet w/o RCs and PhiGnet w/o EVCs, it is better to include another baseline that does not use both of them but ESM-1b, this baseline can clearly explain the important features.
3. There are some other protein function prediction methods that also merge evolutionary information, including some of them for comparison may give a more convincing experimental proof.
4. There some sentences are not necessary too long to understand. For example, Line 307-310, Line 348-351. Please fix/rephrase these alike too long statements.

Reviewer #2:

Remarks to the Author:

The manuscript presents a deep learning method to predict protein functions such as EC numbers and gene ontology identifiers (in categories molecular function, biological process and cellular compartment) from the protein sequence. The method first builds an MSA and predicts evolutionary couplings between each pair of residues (EVC), and, by clustering these, obtained "residue communities" (RCs). Two adjacency matrices are computed from the evolutionary couplings and from the residue communities.

PhiGnet also runs the protein language model ESM-1b to obtain a residue-wise embedding of the sequence. The residue-wise embeddings are the input to two side-by-side graph convolutional networks (GCNs) each using one of the two adjacency matrices as graph. The output of both GCNs is fed into a block of FC layers, ending in a softmax output to predict one function for the protein.

The method is somewhat similar to DeepFri. However, unlike DeepFri it does not require any structural information, which could make it faster to apply to proteins for which not alphafold2 model is available. Also, its idea to include evolutionary couplings extracted from the MSA built for the query protein is quite original. Like DeepFri, PhiGnet is also able to predict the residues in the protein that are most likely to confer the particular function.

The method is compared to a six other tools for function prediction (mostly in Fig. 5), among them those considered current state of the art. PhiGnet performs very favorably on these benchmark comparisons. A method with such improved function prediction performance could be highly impactful.

Major issues:

1. The manuscript is badly written and therefore quite tiresome to read, due to a high density of grammatical and semantic errors, a chaotic structure with often inappropriate order of exposition, redundancies, many omissions, and a long-winded style. This is a shame for such a seemingly powerful method. The manuscript needs complete rewriting to meet the standards expected of a high-impact journal such as Nature Communications. I can only list a few examples here.

* The main part of the manuscript is not at all self-contained and only comprehensible after reading through the Methods section at the end.

* Example of wrong order: Figure 1 shows a plot with an activation score but this score is only mentioned at the end of the results section 6 pages further down. The score is never explained in the main part of the manuscript.

* Examples of missing information: Are 4 different networks trained, one for each functional category (EC, GO MB, GO BP, GO CC)? Why not use weight sharing and train them all at once to improve generalization? Lines 320-324: Where are the data set of functionally annotated proteins taken from?

* Examples of extreme long-windedness and hard-to-digest text": paragraph on lines 187-212, and lines 299-302: "The primary success of our approach, using physics-informed graph convolutional neural networks to adopt hierarchical learning over evolutionary data from massive sequences, over existing both supervised and unsupervised methods significantly improves our efforts in guiding biological/clinical experiments in the future, essentially being implemented to quantitatively identify functional sites at the residue level that would be most important to investigate disease-related variants."

* Example of redundancy that can be omitted: lines 271-275 just reiterate the entire procedure explained in detail before.

* Examples of grammatical and semantic errors: "deep learning over evolutionary information"; "structures of proteins are far less (rarely) than sequences available"; "" sparsely distributed residues"; "energetically ranking functional sites"; "residues at functional sites re recorded from natural evolution"; "resulting from that the"; "Our model substantially presented excellent predictions"; "can accurately characterizes"; state-of-the-arts", "noises", "the remain approaches"; "BLAST and DeepGO were irrespective of whether the changes in sequence identity to the training set"; "GO terms, it could be"; "it could be resulted from"; which might prove biological and clinical research, and for drug discovery."

* Example for redundancy (there are many many more)": Line 171: "Are the computational predictions as accurate as experimentally determined annotations..."? This same question is asked again in another paragraph, Line 269: "We investigated whether the PhiGnet predictions are as accurate as experimental determinations for function annotations of uncharacterized proteins."

2. The new method uses EVC an RCs from an MSA. How does the performance of the method depend on the diversity of the MSAs?

3a. The authors emphasize that the EVCs and RCs are critical for the superior performance of their method. To back up this claim and understand better the sources of improvements, the authors should add as negative control a PhiGnet version that uses nonsensical EVC and RC networks, by inverting the residue indices $i \Rightarrow L + 1 - i$ (where L is the number of residues). This inversion preserves the in- and out-degrees of the original networks.

3b. In Figure 2, add the results of the full PhiGnet method using both EVCs and RCs. Does it really help to use both sources of information rather than using just RCs?

4. The visualizations need to be improved.

* The cord diagrams are a nice negative example for how not to present your data. They cost me minutes to understand what was plotted, because chord diagrams are very unintuitive and hard to read for the data the authors intend to visualize. Chord diagrams are good for showing flows, but there are no flows shown here. It does not become clear that every node is shown twice, for instance, and it does not help that the order is different. The caption gives no guidance on how to interpret these diagrams.

* Figure 1c is hard to read. The upper part is not aligned correctly with the lower part, which can not be spotted easily because the residue numbers below the upper plot are too small to be readable without zooming in.

* Activation score plots: The functional sites annotated in BioLip are indicated by gray traces. These are hard to read. It might be easier to read using grey shading to mark them.

5. Some of the state of the art methods have been trained with less data than phiGnet, which could put them at a disadvantage, couldn't it? How did you ensure that these methods were not trained on your test data?

Minor issues:

* How are the 9 proteins in Figure 4 chosen? How can we know that they are not cherry-picked?

* Are the distributions in Figure 2c (indicated by the whiskers and boxes) over the functional categories?

* What is the impact of predicting only a single function (per category) while we know that many proteins have more than one label per category?

* "We investigated whether the PhiGnet predictions are as accurate as experimental determinations for function annotations of uncharacterized proteins." This question cannot be answered since you have no independent gold standard to compare the accuracy of experimental and in-silico predictions with. This question is therefore misleading and the discussions referring to it have to be adjusted.

* "We applied normalization over all the computed EVCs using a threshold of 0.2 to improve the quality of the EVCs." Unclear.

* grad-CAM approach: Did you also try more standard approaches such as SHAP or integrated gradients?

* Suppl Mat. page 9/14: Fig. S 9 => Fig. S 10

* Fig. S11: What does this plot show? How are the positions defined? What does each point represent?

Reviewer #3:

Remarks to the Author:

This paper proposes PhiGnet, a protein function prediction method that solely relies on sequence information. The experiments in the paper demonstrate that PhiGnet can bridge the gap between sequence and function, even in the absence of structural data. PhiGnet achieves state-of-the-art performance in residue-level protein function annotation by training a physics-informed learning model based on evolutionary data. Additionally, PhiGnet assigns importance scores to each residue for different specific functions, enabling the identification of functional sites at the residue level and providing valuable support for protein characterization and exploration of new functionalities in research and medicine. The paper is logically sound and well-written. It is worth noting that the paper extensively analyzes the prediction performance of PhiGnet for proteins of different sizes, folds, and functions. These analyses provide compelling evidence that the activation scores inferred by PhiGnet accurately represent the residues on the binding interface without requiring any structural information. There are several suggestions to further enhance the paper.

1. The main highlight of the paper is the use of sequence-only information for functional prediction, emphasizing residue-level functional sites. Interestingly, a paper published in 2023 in BIB, SPROF-GO, presents a similar approach: "Here, we propose SPROF-GO, a Sequence-based alignment-free PROtein Function predictor, which leverages a pretrained language model to efficiently extract informative sequence embeddings and employs self-attention pooling to focus on important residues." "Our method was also demonstrated to generalize well on non-homologous proteins and unseen species. Finally, visualization based on the attention mechanism indicated that SPROF-GO is able to capture sequence domains useful for function prediction." Therefore, SPROF-GO should be considered as an important baseline due to its high relevance to the tasks addressed in this paper. Another recent method, ATGO+ (PLOS Computational Biology) should also be used as baseline.

2. The repetition of experiments regarding "PhiGnet learns evolutionary signatures for identification of protein functional sites" and "PhiGnet annotates protein function at the residue level" seems excessive, as they appear in Figure 1b, Figure 1c, Figure 3, and Figure 4. And it's hard to see any essential difference in the things these groups of experiments want to explain. It would be beneficial to streamline and merge these experiments or move some of them to supplementary materials.

3. Regarding the ablation experiment "PhiGnet driven by evolutionary signatures," I suggest moving this section further down in the Results section. Furthermore, it would be informative to include the complete PhiGnet as a baseline in this ablation experiment to visualize the specific improvements brought by each component.

4. The initial paragraph in the Results section, starting with "In the present study, we develop..." lacks a major heading, similar to "PhiGnet annotates protein function at the residue level."

Overall, I believe that some expressions and experimental designs in the paper are redundant and could be organized more effectively.

Response to Reviewer #1:

This paper introduces a physics (evolutionary) information guided deep learning method for protein function prediction. There are two key novel points of this manuscript, the first one is the leverage of evolutionary couplings and residues communities, and the other is the function prediction at the residue level. Besides, this deep method does not require structure data for function prediction, and it outperforms other representative methods by a large margin with better flexibility. The ablation study also proves the effectiveness and rationality of the usage of evolutionary couplings and residues communities. Overall, this paper is interesting and gives some new insights for protein function prediction. Here are some comments for the authors:

1. It is well known that pre-trained models embody refined functional information for protein function prediction, the proposed PhiGnet uses the ESM-1b to extract the raw sequence data, and feeds it into the prediction model. It is unclear whether other comparing methods also use the ESM-1b as the input of sequence data? If not so, it is unclear whether the improved performance is really from the two evolutionary features or not.

Response:

Recently, the ATGO+ model has been developed to predict Gene Ontology (GO) terms for proteins using the pre-trained ESM-1b model [ref. 49]. The ATGO+ model extracts three embeddings from a given protein sequence, specifically from the 31st, 32nd, and 33rd layers of the ESM-1b model. These embeddings are then inputted into a triplet neural network to predict GO terms. In the revised manuscript, we have compared the performance of our model to that of the ATGO+ model in terms of GO term predictions. The ATGO+ model integrates information from a sequence homology-based model and the ESM-1b embeddings to enhance the prediction of GO terms.

Furthermore, a recent model known as CLEAN [ref. 50] has been developed utilizing sequence embeddings from the ESM-1b model for predicting Enzyme Commission (EC) numbers. Based on the comparisons illustrated in Figure 3, our model, PhiGnet, demonstrates superior performance in estimating EC numbers compared to CLEAN, and in making inferences about GO terms compared to ATGO+. More detailed information can be found in the revised manuscript, highlighted in blue.

2. As to results of PhiGnet w/o RCs and PhiGnet w/o EVCs, it is better to include another baseline that does not use both of them but ESM-1b, this baseline can clearly explain the important features.

Response:

We have trained an alternative PhiGnet model by removing the EVCs, RCS, and all GCN layers, and solely utilizing the embedding from ESM-1b to predict protein function annotations related to EC numbers, as illustrated in Table R1 below. With the inclusion of both EVCs and RCs, the model cannot exhibit good performance in the predictions of EC numbers (Table R1). However, the PhiGnet model, incorporating both EVCs and RCs, surpassed the performance of other methods compared, as illustrated in Figure 3 for both EC number and GO term predictions.

Accordingly, these results highlight that, in contrast to using EVCs or RCs individually, the combination of EVCs and RCs leads to enhanced performance of the PhiGnet model in predicting functional annotations. For additional details, please refer to the revised manuscript where specific information is highlighted in blue.

Table R1 Comparison among DeepGO, DeepFRI and PhiGnet (w/ ablations).

Method	Fmax	AUPR
DeepGO	0.3680	0.215
DeepFRI	0.6860	0.695
PhiGnet w/ only ESM-1b	0.3676	0.198
PhiGnet w/o RCs	0.8377	0.854
PhiGnet w/o EVCs	0.8378	0.866
PhiGnet w/ EVCs and RCs	0.8802	0.887

3. There are some other protein function prediction methods that also merge evolutionary information, including some of them for comparison may give a more convincing experimental proof.

Response:

In the revised manuscript, three models utilizing evolutionary information are employed for comparison in predictions of either EC numbers or GO terms. These models include SPROF-GO (which employs the pre-trained protein language model ProtT5-XL-U50) [ref. 48], ATGO+ [ref. 49], and CLEAN [ref. 50]. The predictive performances of these models have been incorporated for comparisons (Figure 3). Additional information can be found in the revised manuscript, specifically in Figure 3 and the Methods section.

4. There some sentences are not necessary too long to understand. For example, Line 307-310, Line 348-351. Please fix/rephrase these alike too long statements.

Response:

We apologize for our imperfect command of written English and have thoroughly reviewed and revised the lengthy statements to ensure clarity with the help of native speakers. Please find more details in the revised manuscript.

Response to Reviewer #2:

The manuscript presents a deep learning method to predict protein functions such as EC numbers and gene ontology identifiers (in categories molecular function, biological process and cellular compartment) from the protein sequence. The method first builds an MSA and predicts evolutionary couplings between each pair of residues (EVC), and, by clustering these, obtained "residue communities" (RCs). Two adjacency matrices are computed from the evolutionary couplings and from the residue communities.

PhiGnet also runs the protein language model ESM-1b to obtain a residue-wise embedding of the sequence. The residue-wise embeddings are the input to two side-by-side graph convolutional networks (GCNs) each using one of the two adjacency matrices as graph. The output of both GCNs is fed into a block of FC layers, ending in a softmax output to predict one function for the protein.

The method is somewhat similar to DeepFri. However, unlike DeepFri it does not require any structural information, which could make it faster to apply to proteins for which not alphafold2 model is available. Also, its idea to include evolutionary couplings extracted from the MSA built for the query protein is quite original. Like DeepFri, PhiGnet is also able to predict the residues in the protein that are most likely to confer the particular function.

The method is compared to a six other tools for function prediction (mostly in Fig. 5), among them those considered current state of the art. PhiGnet performs very favorably on these benchmark comparisons. A method with such improved function prediction performance could be highly impactful.

Major issues:

1. The manuscript is badly written and therefore quite tiresome to read, due to a high density of grammatical and semantic errors, a chaotic structure with often inappropriate order of exposition, redundancies, many omissions, and a long-winded style. This is a shame for such a seemingly powerful method. The manuscript needs complete rewriting to meet the standards expected of a high-impact journal such as Nature Communications. I can only list a few examples here.

Response:

We apologize for our incomplete command of written English and have rewritten the manuscript with the help of native speakers for more clarity. For further details, please refer to the revised manuscript.

* The main part of the manuscript is not at all self-contained and only comprehensible after reading through the Methods section at the end.

Response:

We have revised the manuscript to ensure that it is self-contained without having to browse through the methods section. Please find more details in the revised manuscript.

* Example of wrong order: Figure 1 shows a plot with an activation score but this score is only mentioned at the end of the results section 6 pages further down. The score is never explained in the main part of the manuscript.

Response:

Thank you for your suggestion. We have added a description for the activation score in the main text, as you suggested. Additionally, we have made extensive revisions to the manuscript to enhance clarity.

* Examples of missing information: Are 4 different networks trained, one for each functional category (EC, GO MB, GO BP, GO CC)? Why not use weight sharing and train them all at once to improve generalization? Lines 320-324: Where are the data set of functionally annotated proteins taken from?

Response:

In this study, we trained four models to predict functional categories, which include EC numbers, GO BP, GO CC, and GO MF. We didn't train the PhiGnet model using shared weights for three reasons: (1) the systems of EC numbers and GO terms differ in annotating protein functions. GO terms are to describe gene products, while EC numbers are a hierarchical classification scheme for enzymes based on the reaction catalyzed; (2) the numbers of labels from various categories vary, and this would increase the number of model outputs, leading to heavy computational demands in training and prediction; and (3) the predictive performance of the PhiGnet model would become highly dependent on the fully-connected layers for function annotation prediction if shared weights were used, potentially weakening its performance.

The data sets, including training, validation, and test sets, were obtained from the Protein Data Bank using the same protocol as that of DeepFRI. This point has been clarified in the section of Methods.

* Examples of extreme long-windedness and hard-to-digest text": paragraph on lines 187-212, and lines 299-302: "The primary success of our approach, using physics-informed graph convolutional neural networks to adopt hierarchical learning over evolutionary data from massive sequences, over existing both supervised and unsupervised methods significantly improves our efforts in guiding biological/clinical experiments in the future, essentially being implemented to quantitatively identify functional sites at the residue level that would be most important to investigate disease-related variants."

Response:

This very long statement statement has been split to several sentences and revised to

"The primary success of our approach lies in the utilization of physics-informed graph convolutional neural networks to facilitate hierarchical learning over evolutionary data from massive sequences. This approach not only surpasses existing supervised and unsupervised methods significantly but will also help in guiding future biological and clinical experiments. It is essentially designed to quantitatively identify functional sites at the residue level that hold the utmost importance for investigating disease-related variants."

The paragraph on lines 187-212 has been rewritten for clarity, and now the statements are on lines 105-131 in the revised manuscript.

* Example of redundancy that can be omitted: lines 271-275 just reiterate the entire procedure explained in detail before.

Response:

Thank you for pointing out this redundancy. We have rewritten the paragraph (now on lines 217-231) for clarity. And have rewritten the text with the help of native speakers.

* Examples of grammatical and semantic errors: "deep learning over evolutionary information"; "structures of proteins are far less (rarely) than sequences available"; "" sparsely distributed residues"; "energetically ranking functional sites"; "residues at functional sites re recorded from natural evolution"; "resulting from that the"; "Our model substantially presented excellent predictions"; "can accurately characterizes"; "state-of-the-arts", "noises", "the remain approaches"; "BLAST and DeepGO were irrespective of whether the changes in sequence identity to the training set"; "GO terms, it could be"; "it could be resulted from"; which might prove biological and clinical research, and for drug discovery."

Response:

Thank you for pointing at these mistakes and apologies for our incomplete command of written English. We have corrected the grammatical and semantic errors that were pointed out, and we have also thoroughly reviewed and revised the manuscript accordingly.

1. "deep learning over evolutionary information" has been revised to →

- "applying deep learning to evolutionary data" (Line 14).
- 2. "structures of proteins are far less (rarely) than sequences available" →
The related statements have been rewritten for clarity (Lines 43-45).
- 3. "sparely distributed residues" → "sparsely distributed residues"
- 4. "energetically ranking functional sites" →
The related statements have been rewritten for clarity (Lines 68-69).
- 5. "residues at functional sites re recorded from natural evolution" →
"residues at functional sites are conserved through natural evolution"
- 6. "resulting from that the" → The related statements have been removed for clarity.
- 7. "Our model substantially presented excellent predictions" →
The related statements have been rewritten for clarity (Lines 100-102).
- 8. "can accurately characterizes" → "can accurately characterize".
The related statements have been rewritten for clarity (Lines 126-131).
- 9. "state-of-the-arts" → "state-of-the-art"
- 10. "noises" → "noise"
- 11. "the remain approaches" → "the remaining approaches"
- 12. "BLAST and DeepGO were irrespective of whether the changes in sequence identity to the training set" →
The related statements have been rewritten for clarity (Lines 160-172).
- 13. "GO terms, it could be" → The related statements have been removed for clarity.
- 14. "it could be resulted from" → The related statements have been removed and rewritten for clarity.
- 15. "which might prove biological and clinical research, and for drug discovery." → The related statements have been rewritten for clarity. (Lines 234-236)

Please find more details in the revised manuscript.

* Example for redundancy (there are many many more)": Line 171: "Are the computational predictions as accurate as experimentally determined annotations..."? This same question is asked again in another paragraph, Line 269: "We investigated whether the PhiGnet predictions are as accurate as experimental determinations for function annotations of uncharacterized proteins."

Response:

We have thoroughly reviewed and revised the manuscript accordingly. Please find more details in lines 92-93 and 217-218 in the revised manuscript.

2. The new method uses EVC an RCs from an MSA. How does the performance of the method depend on the diversity of the MSAs?

Response:

We computed the quality of the multiple sequence alignments (MSAs) for the proteins used in both the training and test sets. The distributions of MSAs' quality are illustrated in Figure S15 of the Supplementary Materials. An MSA with a quality score greater than 1 indicates sufficient diversity of sequences, while a score below 1 implies inadequacy for extracting evolutionary data. In this study, the developed model is constructed based on knowledge extracted from these MSAs. The training of the model achieved a good performance by balancing the information derived from MSAs with quality scores less than 1 (it may be a constant bias), as the distributions of MSAs' quality are similar in both the training and test sets. We appreciate the insights gained from this for potential enhancements in future model development.

3a. The authors emphasize that the EVCs and RCs are critical for the superior performance of their method. To back up this claim and understand better the sources of improvements, the authors should add as negative control a PhiGnet version that uses nonsensical EVC and RC networks, by inverting the residue indices $i \Rightarrow L + 1 - i$ (where L is the number of residues). This inversion preserves the in- and out-degrees of the original networks.

Response:

As you suggested, we carried out the training and prediction on the same dataset. Subsequently, we compared the performance between the models of EVCs & RCs and of inverted EVCs and RCs as illustrated in Figure R1. It is surprising that the models achieved comparable precision-recall curve and Fmax scores, indicating the deep learning model has the ability to bias the inversions for target outputs. All indices of residues have been inverted in the same way ($i \Rightarrow L+1-i$), but the target annotations are not changed. In this way, the model may learn the inversions as a constant bias in the training and prediction. We conclude that matrix inversion cannot be used as a negative control, and that we have to rely on the ablation experiments to demonstrate the importance of ECs and RCs.

Fig. R1. Comparison between PhiGnet and PhiGnet w/ inverted EVCs & RCs.

3b. In Figure 2, add the results of the full PhiGnet method using both EVCs and RCs. Does it really help to use both sources of information rather than using just RCs?

Response:

Yes, when compared to using just RCs/EVCs, the performance of the PhiGnet model was improved by the utilization of both EVCs and RCs. The predictive performance of PhiGnet using both EVCs and RCs has been illustrated and compared to other methods in Figure 3 of the revised manuscript. For clarity, the results of PhiGnet using both EVCs and RCs are also presented in Figures S9 and S10 of the Supplementary Materials. Further details can be found in the revised manuscript and its Supplementary Materials.

4. The visualizations need to be improved.

* The cord diagrams are a nice negative example for how not to present your data. They cost me minutes the understand what was plotted, because chord diagrams are very unintuitive and hard to read for the data the authors intend to visualize. Cord diagrams are good for showing flows, but there are no flows shown here. It does not become clear that every node is shown twice, for instance, and it does not help that the order is different. The caption gives no guidance on how to interpret these diagrams.

Response:

The residue communities were identified from residues exhibiting high evolutionary couplings. For these residues, we computed their conservation scores represented by the green bars in Figure 1b. Subsequently, the residues were classified into two groups: Community I (highlighted in red) and Community II (highlighted in blue). The strength of coupling for each residue is illustrated using a bar (in either red or blue), with darker colors indicating stronger coupling with its partners. In the revised manuscript, labels for the residue communities have been included in Figure 1b. Additionally, we have revised the caption to improve clarity. For more details, please refer to the revised manuscript.

* Figure 1c is hard to read. The upper part is not aligned correctly with the lower part, which can not be spotted easily because the residue numbers below the upper plot are too small to be readable without zooming in.

Response:

We have enhanced the visualization of conservation (the sequence logo, as illustrated in Figure 1c, upper panel) for the MglA protein. Due to the density in the graphical representation of amino acids at each site, we cropped the fragments that encompass the residues identified by the BiLip data. The coordinates on the x-axis of the upper graph are identical to those of the lower graph. Please refer to the improved visualization of Figure 1c in the revised manuscript.

* Activation score plots: The functional sites annotated in BioLip are indicated by gray traces. These are hard to read. It might be easier to read using grey shading to mark them.

Response:

Thank you for your suggestion. The functional sites annotated in BioLip have been indicated by the symbol of 'y' for clarity. Please find the details in Figure 1 and Figure 2 of the revised manuscript and Figure S1 of the revised supplementary materials.

5. Some of the state of the art methods have been trained with less data than phiGnet, which could put them at a disadvantage, couldn't it? How did you ensure that these methods were not trained on your test data?

Response:

Our model was trained using the dataset that was collected by the same protocol as used in DeepFRI. The compared methods, such as DeepGO and ProtelInfer, utilized in this study, were fine-tuned on their respective datasets. For the comparison, we employed pre-trained models of various methods to predict function annotations on the same test dataset. Throughout this study, we didn't carry out re-training of the supervised methods for function annotation prediction, even though the datasets used in different methods might partially overlap. We conducted performance comparisons between our model and these methods with their fine-tuned parameters, as asserted in the corresponding references.

Minor issues:

* How are the 9 proteins in Figure 4 chosen? How can we know that they are not cherry-picked?

Response:

We screened about 5,000 proteins in the test sets of GO terms and EC numbers using three rules: (1) a protein has a crystal structure; (2) the protein exhibits either ligand, ion, or DNA binding; and (3) the binding sites are also recorded in the BioLip database. Of these proteins, we focused on thirty examples to demonstrate the mapped activation scores on their tertiary structures (9 proteins in Figure 2 and 21 proteins in Figure S1 of the Supplementary Materials).

* Are the distributions in Figure 2c (indicated by the whiskers and boxes) over the functional categories?

Response:

The distributions are from the same functional category (e.g., EC number, as shown in Figure 3c) but with different thresholds for sequence identity between proteins in the test and training sets. For instance, in the test set, we selected all the proteins with a sequence identity lower than a specified maximum value (e.g., 50%) in the EC number category, and we computed the Fmax scores for these proteins. These calculations resulted in a distribution of Fmax scores for the proteins.

* What is the impact of predicting only a single function(per category) while we know that many proteins have more than one label per category?

Response:

It appears that the reviewer thinks that our model only outputs a single functional annotation. We apologize for the lack of clarity. In the present study, we developed the PhiGnet model as a multi-class predictor for protein function annotations (both EC numbers and GO terms). The model predicts multiple annotations for each protein. For each function category, the model has an output vector that contains probabilities. The higher the probability is, the more confidence we have to assign the annotation to the proteins. Proteins can have multiple annotations of the same functional category, provided that their probability is high enough.

We don't train the model as a single-class/label predictor, as assigning a single function to a protein will increase the number of samples in both training and test datasets. For example, protein A has two function annotations, while proteinB has three unique annotations that are different from that of protein A. Subsequently, we get six samples although there are only two proteins. This leads to extremely heavy computations for the task in training a single-label predictor.

To simplify the computations of single-label predictor, we extracted every single label of each protein and computed the Fmax scores, area under precision-recall (AUPR), and Matthew's correlation coefficient (MCC) to assess the performance of PhiGnet for proteins of a single function (Table R2).

Table R2 PhiGnet's performance with respect to a single function per protein.

Category	Fmax	AUPR	MCC
EC number	0.9535	0.9293	0.8811

BP	0.9159	0.9434	0.6438
CC	0.9362	0.9699	0.8324
MF	0.9450	0.9441	0.7990

* "We investigated whether the PhiGnet predictions are as accurate as experimental determinations for function annotations of uncharacterized proteins." This question cannot be answered since you have no independent gold standard to compare the accuracy of experimental and in-silico predictions with. This question is therefore misleading and the discussions referring to it have to be adjusted.

Response:

Indeed, thank you for pointing this out. We have made the necessary correction to the statement as it presents on lines 217-231 in the revised manuscript.

* "We applied normalization over all the computed EVCs using a threshold of 0.2 to improve the quality of the EVCs." Unclear.

Response:

The threshold was employed to mitigate potential noise arising from coevolution or weak couplings between pairwise residues. To obtain the threshold, we varied its values that were applied to the normalized EVCs/RCs and compared the model's performances with different values of the threshold (see Figure S8 of the Supplementary Materials). Among our preliminary demonstrations, the PhiGnet model exhibited the best performance with a threshold of 0.2. In all the training and tests, we chose the threshold of 0.2 for both EVCs and RCs. Further details can be found in the revised manuscript and its Supplementary Materials.

* grad-CAM approach: Did you also try more standard approaches such as SHAP or integrated gradients?

Response:

Thank you for your suggestions. In the present model, we have employed the grad-CAM method for the calculations, while SHAP or integrated gradients have not been used. We appreciate your suggestion and will consider incorporating SHAP or integrated gradients in future model developments.

* Suppl Mat. page 9/14: Fig. S 9 => Fig. S 10

Response:

We have made the revision in the revised Suppl Mat..

* Fig. S11: What does this plot show? How are the positions defined? What does each point represent?

Response:

Figure S11 was generated using the t-distributed stochastic neighbor embedding (t-SNE) method. We first computed the Hamming distance between protein sequences as a measure of similarity for clustering purposes. The t-SNE tool was employed to visualize the similarity of the proteins. The positions of the clusters were determined from the two components of the t-SNE outputs and are depicted in Figure S11. Each point in Figure S11 represents a cluster, with the size of each bubble indicating the cluster's size.

Response to Reviewer #3:

This paper proposes PhiGnet, a protein function prediction method that solely relies on sequence information. The experiments in the paper demonstrate that PhiGnet can bridge the gap between sequence and function, even in the absence of structural data. PhiGnet achieves state-of-the-art performance in residue-level protein function annotation by training a physics-informed learning model based on evolutionary data. Additionally, PhiGnet assigns importance scores to each residue for different specific functions, enabling the identification of functional sites at the residue level and providing valuable support for protein characterization and exploration of new functionalities in research and medicine. The paper is logically sound and well-written. It is worth noting that the paper extensively analyzes the prediction performance of PhiGnet for proteins of different sizes, folds, and functions. These analyses provide compelling evidence that the activation scores inferred by PhiGnet accurately represent the residues on the binding interface without requiring any structural information. There are several suggestions to further enhance the paper.

1. The main highlight of the paper is the use of sequence-only information for functional prediction, emphasizing residue-level functional sites. Interestingly, a paper published in 2023 in BIB, SPROF-GO, presents a similar approach: "Here, we propose SPROF-GO, a Sequence-based alignment-free PROtein Function predictor, which leverages a pretrained language model to efficiently extract informative sequence embeddings and employs self-attention pooling to focus on important residues." "Our method was also demonstrated to generalize well on non-homologous proteins and unseen species. Finally, visualization based on the attention mechanism indicated that SPROF-GO is able to capture sequence domains useful for function prediction." Therefore, SPROF-GO should be considered as an important baseline due to its high relevance to the tasks addressed in this paper. Another recent method, ATGO+ (PLOS Computational Biology) should also be used as baseline.

Response:

Thank you for your suggestion. In the revised paper, we have employed the SPROF-GO [ref. 48] and ATGO+ [ref. 49] models for comparing the predictions of GO terms. Moreover, we have compared the developed model to the CLEAN model [ref. 50] on the predictions of the EC numbers. These comparisons are illustrated in Figure 3. For additional information, please refer to the revised manuscript where relevant details are highlighted in blue.

2. The repetition of experiments regarding "PhiGnet learns evolutionary signatures for identification of protein functional sites" and "PhiGnet annotates protein function at the residue level" seems excessive, as they appear in Figure 1b, Figure 1c, Figure 3, and Figure 4. And it's hard to see any essential difference in the things these groups of experiments want to explain. It would be beneficial to streamline and merge these experiments or move some of them to supplementary materials.

Response:

Thank you for your suggestions. In the revised manuscript, we have relocated Figure 3 (now Figure 4 in the revised manuscript) as part of the ablation analysis. In this ablation, we have elucidated the contributions of residue communities to the PhiGnet model in identifying the residues at functional sites. These insights provide additional context for interpreting the outcomes presented in Figure 1b, 1c, and the newly revised Figure 2 (Figure 4 previously). Please find more details in the revised manuscript.

3. Regarding the ablation experiment "PhiGnet driven by evolutionary signatures," I suggest moving this section further down in the Results section. Furthermore, it would be informative to include the complete PhiGnet as a baseline in this ablation experiment to visualize the specific improvements brought by each component.

Response:

Thank you for your suggestions. In the revised manuscript, we have moved the ablation experiment titled 'PhiGnet driven by evolutionary signatures' to a position after the section titled 'PhiGnet outperforms other state-of-the-art methods' in the Results.

As per your suggestion, we have included the results of the complete PhiGnet (with both EVCs and RCs) for the ablation experiments (Figure S9 and S10). Further details can be found in the revised manuscript and its Supplementary Materials.

4. The initial paragraph in the Results section, starting with "In the present study, we develop..." lacks a major heading, similar to "PhiGnet annotates protein function at the residue level."

Response:

Thank you for your suggestion. We have provided the sub-section with a title of "The PhiGnet model for protein function annotations".

Overall, I believe that some expressions and experimental designs in the paper are redundant and could be organized more effectively.

Response:

Thank you for your suggestions. We have revised the manuscript by eliminating redundant descriptions and enhancing its organization as per your suggestions. For additional information, please refer to the revised manuscript, where pertinent details are highlighted in blue.

Reviewers' Comments:

Reviewer #1:

Remarks to the Author:

I am satisfied with the authors' revisions and efforts on improving the manuscript to clarify concerns of reviewers.

Reviewer #2:

Remarks to the Author:

The manuscript has improved a lot, both by improving the text and by adding analyses suggested by the three reviewers.

Two major points still need to be addressed.

1. The fact that nonsensical graphs, obtained by reverting the indices of the amino acid residues ($i \Rightarrow L+1-i$), is very interesting and also worrisome, because this negative control appears to clearly show that the graphs obtained from EVDs and RCs do not have an influence on the performance of the model. How is it possible that the model keeps its performance when the network connections in the residue order-inverted EVD and RC graphs do not connect coupled residues or residue communities?

The authors write that "Through precision and robustness comparisons, we have demonstrated that the evolutionary signatures (EVCs and RCs) constitute crucial attributes capable of enhancing deep learning-based methods for protein function annotations." This is obviously not true given the results of the negative control experiment.

Instead of simply omitting this troublesome observation from the manuscript, it should be further investigated. What is the performance of the model if residue indices are subjected to a random permutation? What is the performance if random networks are generated by scrambling the EVD network such that in- and outdegrees are conserved (and the RC network is computed from the randomized EVD network using the same approach as for the non-randomized EVD network)? The interpretation of the results of these analyses should be discussed.

2. Overfitting on the training data is a pervasive problem in deep learning that leads to many overconfident results. Any data leakage should therefore be carefully avoided. To test the influence of the similarity of training proteins on prediction results, they filter training proteins by maximum sequence identity to the closest training protein. However, this means that validation proteins can be nearly identical to proteins in the validation set. The filtering should be done against proteins in the training and validation set.

Minor:

3. "FC block" in Fig. 1a is never defined.

Reviewer #3:

Remarks to the Author:

The authors have addressed most of the major issues raised by the reviewers. They have added new comparisons with related methods such as SPROF-GO, ATGO+, and CLEAN, and shown that PhiGnet outperforms them in predicting EC numbers and GO terms. They have also performed ablation studies to demonstrate the importance of EVC and RC on PhiGnet performance. In

addition, they have improved the visualization and interpretation of the results and corrected many syntactic and semantic errors, among others. However, there are still some issues that need to be addressed.

Major

1. The authors have added some comparisons with more recent methods that use evolutionary information (e.g., SPROF-GO, ATGO+, and CLEAN).

For a fair comparison, the same training and test data should be used. Additionally, the comparisons are only shown in Figure 3 and the results are not discussed or analyzed in detail. For example, why does PhiGnet perform better than these methods? How do the evolutionary features used by each method differ? Methods such as SPROF-GO have a high degree of overlap with the core innovations of this article. These questions should be addressed in the manuscript in order to provide more insight and evidence for the novelty and superiority of PhiGnet.

2. The authors performed ablation experiments to demonstrate the importance of EVC and RC, but the differences between PhiGnet without RC and PhiGnet without EVC were very small and statistically insignificant. Furthermore, the authors do not explain why using only ESM-1b embedding leads to very poor performance, which is surprising considering that ESM-1b is a powerful pre-trained model that captures sequence features.

3. In line 287, "These actions were informed by the experimental design's focus on hyper-parameter optimization through grid search.", how robust is the model for different thresholds for filtering evolutionary coupling and residue communities?

Minor

1. In the caption of figure 3, "(e) Violin plot displaying AUPR (left) and F-max score (right) for different methods in predicting CC, BP, and MF". It seems that the right one is simple line chart not a violin plot. It is suggested that authors clarify it.

2. In line 302-304, there is a duplication here, two identical sentences "In one channel, a channel of stacked GCNs gathers information from the sequence embedding using evolutionarily coupled residues as graph nodes". It is suggested that authors delete one of them.

3. As stated in the revise manuscript, authors add ATGO, SPROF-GO, and CLEAN as competing methods. Therefore, the three methods should be mentioned in the first paragraph of "Comparison with existing approaches" (line 338-340). For example, "In the present study, we compared our PhiGnet model to five methods, including BLAST, FunFams, DeepGO, DeepFRI, and ProteInfer in details" should be revised as "In the present study, we compared our PhiGnet model to eight methods, including BLAST, FunFams, DeepGO, DeepFRI, ProteInfer, ATGO, SPROF-GO, and CLEAN in details".

4. Many of the figures in the paper look strange and their clarity seems to be inconsistent. Take Figure 1. C as an example. The numbers "134", "143", "163", and "170" seem to be obviously distorted. This problem is more obvious in the "GDP" and "SO4" in the figure.

5. Line 71, "exmample" is misspelled. The authors need to proofread the full text carefully.

Response to Reviewer #1:

I am satisfied with the authors' revisions and efforts on improving the manuscript to clarify concerns of reviewers.

Response:

Thank you for your valuable comments and suggestions which have improved our manuscript.

Response to Reviewer #2:

The manuscript has improved a lot, both by improving the text and by adding analyses suggested by the three reviewers.

Two major points still need to be addressed.

1. The fact that nonsensical graphs, obtained by reverting the indices of the amino acid residues ($i \Rightarrow L+1-i$), is very interesting and also worrisome, because this negative control appears to clearly show that the graphs obtained from EVDs and RCs do not have an influence on the performance of the model. How is it possible that the model keeps its performance when the network connections in the residue order-inverted EVD and RC graphs do not connect coupled residues or residue communities?

Response:

In the PhiGnet model, EVCs and RCs are used as graph edges that link the nodes (defined by the sequence embedding). As suggested, we reverted the indices of the residues by $i \Rightarrow L+1-i$ for re-training the model (Fig. R1). The performance of PhiGnet was not affected a lot by the rotation of the ECs and RCs, as the model would learn the bias when they were rotated in training and test for the same target outputs.

Moreover, the swapped "edges" may have the same values. Here we take a matrix of EVCs as an example (Fig. R1), the second and third residues are linked based on the measurement in EVCs (the cell in blue). Their 'edge' is subsequently determined by that between the fifth and sixth residues (the cell in green) when their indices are reverted, vice versa. The edges between graph nodes are changed by reverting the indices, and the pattern of each EVCs/RCs matrix is the same but clockwise rotated 180° . This rotation has been applied to all the proteins, and the PhiGnet model adapts itself to learn the rotated patterns. After re-training, we achieved the similar performance of the PhiGnet model with rotated EVCs and RCs, as no embedding of indices is used in the training and prediction. For example, a deep learning-based model generally cannot recognize a photo of a dog with 180° rotation if it is trained on non-rotated photos. However, if the model is re-trained on all rotated dog photos, it would recognize the rotated photo with similar accuracy. If instead of inverting the indices, we use random indices, then the performance of the model indeed crumbles (see below).

Fig. R1. Clockwise rotation (180°) of EVCs/RCs.

The authors write that "Through precision and robustness comparisons, we have demonstrated that the evolutionary signatures (EVCs and RCs) constitute crucial attributes capable of enhancing deep learning-based methods for protein function annotations." This is obviously not true given the results of the negative control experiment.

Instead of simply omitting this troublesome observation from the manuscript, it should be further investigated. What is the performance of the model if residue indices are subjected to a random permutation? What is the performance if random networks are generated by scrambling the EVD network such that in- and outdegrees are conserved (and the RC network is computed from the randomized EVD network using the same approach as for the non-randomized EVD network)? The interpretation of the results of these analyses should be discussed.

Response:

Indeed, using random values alleviates the problem that inverted patterns can be learnt just as well as non-inverted ones. We have therefore the following controls (1) using a random permutations for residue indices of EVCs and RCs (random permutation for short); (2) EVCs with fully random values, and RCs were derived from that EVCs (random values for short); and (3) set zeros for both EVCs and RCs (zero for short). The results are illustrated in Fig. R2 as below and Fig. S11 of the Supplementary Materials (for the ablation experiments). It is very apparent that the model cannot perform in these conditions Note that the scores of F_{max} and AUPR are not 0 because of the non-zero bias of the model although the weights have been muted by substituting EVCs and RCs with zeros.

Fig. R2. Performance of PhiGnet with different adjacency matrices over the function categories of (a) BP, (b) CC, (c) MF, and (d) EC.

2. Overfitting on the training data is a pervasive problem in deep learning that leads to many overconfident results. Any data leakage should therefore be carefully avoided. To test the influence of the similarity of training proteins on prediction results, they filter training proteins by maximum sequence identity to the closest training protein. However, this means that validation proteins can be nearly identical to proteins in the validation set. The filtering should be done against proteins in the training and validation set.

Response:

To construct the datasets, we carried out the same protocol used in the DeepFRI model. We collected proteins (until 10/2021) and randomly divided them into three subsets for training, validation, and test. The protocol defines 90% for the maximum sequence identity between pairwise sequences. We computed the sequence identity of pairwise proteins within the test dataset and made the distributions of the sequence identity as illustrated in Fig. S17 of the Supplementary Materials. The test/validation proteins are not 'nearly identical to' the proteins in the test/validation set. Most of the proteins are similar to each other with less than 50% sequence identity. Please find more details in the Supplementary Materials.

Minor:

3. "FC block" in Fig. 1a is never defined.

Response:

The phrase "FC blocks" has been replaced by "a block of two fully connected (FC) layers" for clarity in the context highlighted in blue and Fig. 1a of the revised manuscript.

Response to Reviewer #3:

The authors have addressed most of the major issues raised by the reviewers. They have added new comparisons with related methods such as SPROF-GO, ATGO+, and CLEAN, and shown that PhiGnet outperforms them in predicting EC numbers and GO terms. They have also performed ablation studies to demonstrate the importance of EVC and RC on PhiGnet performance. In addition, they have improved the visualization and interpretation of the results and corrected many syntactic and semantic errors, among others. However, there are still some issues that need to be addressed.

Major

1. The authors have added some comparisons with more recent methods that use evolutionary information (e.g., SPROF-GO, ATGO+, and CLEAN).

For a fair comparison, the same training and test data should be used. Additionally, the comparisons are only shown in Figure 3 and the results are not discussed or analyzed in detail. For example, why does PhiGnet perform better than these methods? How do the evolutionary features used by each method differ? Methods such as SPROF-GO have a high degree of overlap with the core innovations of this article. These questions should be addressed in the manuscript in order to provide more insight and evidence for the novelty and superiority of PhiGnet.

Response:

For the testing, we use the same test dataset for comparison between PhiGnet and other methods, including SPROF-GO, ATGO+, and CLEAN, as pointed out by the reviewer. This allows us to assess the performance of various models under identical evaluation conditions.

For the training datasets, re-training each of the models on identical training sets is not feasible with our computing power. Instead, we obtained the pre-trained model of each method (SPROF-GO, ATGO+, CLEAN, and the remaining methods) as presented in the corresponding references, without altering their parameters for the comparisons. Note, however that SPROF-GO, ATGO+ and CLEAN have all been published between 2022 and 2023, and that the size of the Uniprot database, which constitute the basis of all training datasets, has not changed substantially in recent years (see <https://web.expasy.org/docs/relnotes/relstat.html>, ~570k sequences nowadays against ~540k 10 years ago). To train the PhiGnet model, we collected ~42k and ~20k proteins with GO terms and EC numbers using the very same protocol as that of the DeepFRI model. The SPROF-GO model uses ~50k, ~83k, and ~74k proteins of MF, BP, and CC, respectively, for training its parameters. ATGO+ and CLEAN were trained by ~122k and ~220k proteins, respectively. It is therefore highly unlikely that superior performance of our model is due to a better training dataset since, 1-the Uniprot hasn't changed much in the past years, 2-we used on average less proteins with GO or EC numbers than the recent competing models. The major improvement of PhiGnet instead comes from the knowledge (EVCs and RCs) derived by the Potts model and the spectrum analysis. The model uses not only the sequence embedding from the pre-trained ESM-1b model but also the derived knowledge, and this combination results in improvement in its predictive performance.

In the PhiGnet model, we use the sequence embedding as graph nodes and the derived knowledge (EVCs and RCs) as graph edges to model the relationship between protein sequence and its function annotation(s). Different from PhiGnet, the compared methods extract homologous information (e.g., BLAST, FunFams, and Pannzer), or use sequence embedding from the evolutionary data (e.g., SPROF-GO), or extract features (e.g., ATGO+), or use sequence embedding for computing similarity (e.g., CLEAN), to predict function annotations for proteins.

The main differences among the methods (SPROF-GO, ATGO+, CLEAN, and PhiGnet) have been presented in Table R1. As illustrated in Table R1, there is no overlap between PhiGnet and SPROF-GO (except their input and output). We have also added more details to discuss the comparison in the revised manuscript.

Table R1. Differences among SPROF-GO, ATGO+, CLEAN, and PhiGnet.

	SPROF-GO	ATGO+	CLEAN	PhiGnet
Input	Sequence (max. 2,000 AA)	Sequence (max. 10,000 AA*)	Sequence (max. 1,022 AA)	Sequence (max. 1024 AA)
Embedding	ProtT5-XL-U50	ESM-1b	ESM-1b	ESM-1b
Physics-informed	No	No	No	Yes
With homology	Yes	Yes	No	No
Model	Three multilayer perceptrons (MLPs) and self-attention pooling. Two parallel MLPs to learn an attention vector and a hidden embedding matrix, fused by the third MLP for predicting annotations.	Four fully connected (FC) networks. Three parallel FC networks for feature extraction, and a FC network for fusion of features for predicting annotations.	Based on a contrastive learning model for predicting annotations.	Two groups of stacked graph convolutional networks (GCN) and a fully connected (FC) block of two dense layers. The GCN layers learn sequence embedding with physical constraints of EVCs and RCs at two levels. The FC block is to present confidence for annotations.
Output	BP, CC, MF	BP, CC, MF	EC number	BP, CC, MF, EC number

*: collected from the ATGO+ webserver

AA: amino acid
BP: biological process
CC: cellular component
MF: molecular function
EC number: enzyme commission number

Please find more details in the revised manuscript.

2. The authors performed ablation experiments to demonstrate the importance of EVC and RC, but the differences between PhiGnet without RC and PhiGnet without EVC were very small and statistically insignificant. Furthermore, the authors do not explain why using only ESM-1b embedding leads to very poor performance, which is surprising considering that ESM-1b is a powerful pre-trained model that captures sequence features.

Response:

There are two potential factors that lead to the small difference between PhiGnet without RCs and PhiGnet without EVCs. One is that RCs and EVCs are not independent. Indeed, RCs are derived from EVCs as highly-ordered patterns of interactions between pairwise residues. It is therefore not entirely surprising that the increase in performance of both is less than additive. The similarity between the EVCs and RCs would lead to a similar performance increase of the model. The other factor is that we altered the PhiGnet model and trained each using a single group of stacked graph convolutional networks. This alteration would lead the model to learn information for target function annotation(s) using the shared data (graph edges that are defined by EVCs or RCs). Together, the two factors may lead to the similar performance between PhiGnet without RCs and PhiGnet without EVCs.

Yes, the pre-trained ESM-1b model is good at capturing sequence features, and that is why we use its embedding for predictions of protein function annotations. But two fully connected (FC) layers in PhiGnet (see Supplementary Materials) are not specially designed based on the ESM-1b model. The FC layers have been devised to process the extracted data from the graph networks for the classification of function annotations. In the ablation of removing both EVCs and RCs, the FC layers cannot well learn the ESM-1b embedding for good performance over the prediction of protein function annotations. It may require much more FC layers to present better performance, but it is out of the range of this study.

3. In line 287, "These actions were informed by the experimental design's focus on hyper-parameter optimization through grid search.", how robust is the model for different thresholds for filtering evolutionary coupling and residue communities?

Response:

The relationship between the thresholds and the model's performance has been present in Fig. S8 of the Supplementary Materials (revision V1). We have made an indication in the line 308 for clarity. Please find more details in the Supplementary Materials.

Minor

1. In the caption of figure 3, "(e) Violin plot displaying AUPR (left) and F-max score (right) for different methods in predicting CC, BP, and MF". It seems that the right one is simple line chart not a violin plot. It is suggested that authors clarify it.

Response:

We have made revision of the description for clarity. Please find more details highlighted in blue in the revised manuscript.

2. In line 302-304, there is a duplication here, two identical sentences "In one channel, a channel of stacked GCNs gathers information from the sequence embedding using evolutionarily coupled residues as graph nodes". It is suggested that authors delete one of them.

Response:

Thank you for your suggestion. We have removed the duplication in the revised manuscript.

3. As stated in the revise manuscript, authors add ATGO, SPROF-GO, and CLEAN as competing methods. Therefore, the three methods should be mentioned in the first paragraph of "Comparison with existing approaches" (line 338-340). For example, "In the present study, we compared our PhiGnet model to five methods, including BLAST, FunFams, DeepGO, DeepFRI, and ProtelInfer in details" should be revised as "In the present study, we compared our PhiGnet model to eight methods, including BLAST, FunFams, DeepGO, DeepFRI, ProtelInfer, ATGO, SPROF-GO, and CLEAN in details".

Response:

Thank you for your suggestions. We have made the revision highlighted in blue in the revised manuscript.

4. Many of the figures in the paper look strange and their clarity seems to be inconsistent. Take Figure 1. C as an example. The numbers "134", "143", "163", and "170" seem to be obviously distorted. This problem is more obvious in the "GDP" and "SO4" in the figure.

Response:

The sequence logo representation in Fig. 1c was cropped based on the functional sites. The indices are to indicate the positions of starting and ending for the cropped fragments.

GDP is an abbreviation of guanosine di-nucleotide, while SO₄ is a sulfate ion.

5. Line 71, "exmaple" is misspelled. The authors need to proofread the full text carefully.

Response: The typo has been corrected, we have double-checked the manuscript.

Reviewers' Comments:

Reviewer #2:

Remarks to the Author:

The authors have improved the manuscript a lot.

My main issue has not been addressed, however. This was it:

"1. The fact that nonsensical graphs, obtained by reverting the indices of the amino acid residues ($i \Rightarrow L+1-i$), is very interesting and also worrisome, because this negative control appears to clearly show that the graphs obtained from EVDs and RCs do not have an influence on the performance of the model. How is it possible that the model keeps its performance when the network connections in the residue order-inverted EVD and RC graphs do not connect coupled residues or residue communities?"

The authors' response shows that they have obviously misunderstood the suggested test. Of course, when you reverse the order of residue indices for the EVC / RC graphs AND the input embeddings, the network should learn (up to noise) exactly the same as without inversion. The authors correctly compare this with training a neural network with images rotated by 180 degrees. But this is a trivial outcome and not useful.

What I was requesting was to test whether the specific EVC and RC graphs that are determined for each protein are important or not, by inverting the order of residues **for the graphs only**, without also inverting them to the embeddings and thereby defeating the purpose.

The tests that the authors have now done are not useful as the randomizations of residue order are presumably all applied to both embeddings and graphs. This creates embeddings without much meaning as the embeddings are interpreted within the context of the protein. Hence anything but a drastic drop of performance would be surprising at any rate, whether the graphs before randomization were arbitrary or not.

It is quite central to understand the causes for the good performance of PhiGnet. The manuscript emphasizes the importance of being "physics informed", by which I guess the authors mean these two graphs. "The major improvement of PhiGnet instead comes from the knowledge (EVCs and RCs) derived by the Potts model and the spectrum analysis." So it is critical to back up this claim and carefully analyse the causes of improvement, which are unclear at this point. I fully agree with reviewer 3 (issue 1) on this.

Small point: I find the characterization of the method as "physics informed" misleading, as I do not see the physics involved. The term suggests that physical potentials or the like have been used. "Statistics-informed" would be more accurate.

Reviewer #3:

Remarks to the Author:

The authors have addressed some issues we raised last time. Nevertheless, uncertainties persist regarding the superiority of PhiGnet.

1. To thoroughly assess PhiGnet's superiority, it is imperative to employ identical training and test data, facilitating a direct performance comparison with other models—a fundamental principle universally recognized in machine learning. In the realm of protein function prediction, this principle gains heightened significance due to disparate data collection methodologies among various methods. For instance, PhiGnet adopts the DeepFRI protocol, filtering less frequent GO

terms for functional annotations, whereas both SPROF-GO and ATGO+ utilize the CAFA protocol, retaining all GO terms in experimentally verified annotations. Notably, a majority of functional annotations correspond to these less frequent GO terms, underscoring the substantial variance in training data across these methodologies. Presenting the performance of different models without aligning their training data could mislead interpretations.

2. To ascertain the practical applicability and generalizability of PhiGnet, the authors have the opportunity to evaluate its performance using the CAFA5 test data, available at this URL: <https://www.kaggle.com/competitions/cafa-5-protein-function-prediction/overview>. CAFA5, co-organized with the UniProt team, provides the benchmark for this assessment. Recently, the CAFA5 organizers finalized the results and enabled late submissions. The authors can leverage functional annotations up to August 21, 2003, to train PhiGnet and subsequently evaluate its performance on the CAFA5 test data.

Response to Reviewer #2:

The authors have improved the manuscript a lot.

My main issue has not been addressed, however. This was it:

“1. The fact that nonsensical graphs, obtained by reverting the indices of the amino acid residues ($i \Rightarrow L+1-i$), is very interesting and also worrisome, because this negative control appears to clearly show that the graphs obtained from EVDs and RCs do not have an influence on the performance of the model. How is it possible that the model keeps its performance when the network connections in the residue order-inverted EVD and RC graphs do not connect coupled residues or residue communities?”

The authors' response shows that they have obviously misunderstood the suggested test. Of course, when you reverse the order of residue indices for the EVC / RC graphs AND the input embeddings, the network should learn (up to noise) exactly the same as without inversion. The authors correctly compare this with training a neural network with images rotated by 180 degrees. But this is a trivial outcome and not useful.

What I was requesting was to test whether the specific EVC and RC graphs that are determined for each protein are important or not, by inverting the order of residues **for the graphs only**, without also inverting them to the embeddings and thereby defeating the purpose.

The tests that the authors have now done are not useful as the randomizations of residue order are presumably all applied to both embeddings and graphs. This creates embeddings without much meaning as the embeddings are interpreted within the context of the protein. Hence anything but a drastic drop of performance would be surprising at any rate, whether the graphs before randomization were arbitrary or not.

It is quite central to understand the causes for the good performance of PhiGnet. The manuscript emphasizes the importance of being “physics informed”, by which I guess the authors mean these two graphs. “The major improvement of PhiGnet instead comes from the knowledge (EVCs and RCs) derived by the Potts model and the spectrum analysis.” So it is critical to back up this claim and carefully analyse the causes of improvement, which are unclear at this point. I fully agree with reviewer 3 (issue 1) on this.

Response:

We thank the reviewer for clarifying this point that we had indeed missed, and apologize. As suggested, we have re-trained the PhiGnet model with inversions of EVC and RC, and the performance comparison is illustrated in Fig. R1. Retaining the sequence embedding, we conducted various training iterations, including models with inverted EVC, inverted RC, and inversions applied to both.

Inverting either EVC or RC compromised PhiGnet's performance dramatically. Inverting both was even worse. Together, these tests show the importance of both EVCs and RCs in PhiGnet's performance.

Fig. R1. PhiGnet's performance over predictions of EC numbers.

Small point: I find the characterization of the method as “physics informed” misleading, as I do not see the physics involved. The term suggests that physical potentials or the like have been used. “Statistics-informed” would be more accurate.

Response:

Thank you for your suggestion. The term 'physics informed' was used to stress that the EVCs and RCs were derived from a Potts model used in statistical physics. We agree with the reviewer that this is not straightforward and misleading. The term has therefore been updated to 'statistics informed' as proposed by the reviewer. Please refer to the revised manuscript for more details.

Response to Reviewer #2:

The authors have addressed some issues we raised last time. Nevertheless, uncertainties persist regarding the superiority of PhiGnet.

1. To thoroughly assess PhiGnet's superiority, it is imperative to employ identical training and test data, facilitating a direct performance comparison with other models—a fundamental principle universally recognized in machine learning. In the realm of protein function prediction, this principle gains heightened significance due to disparate data collection methodologies among various methods. For instance, PhiGnet adopts the DeepFRI protocol, filtering less frequent GO terms for functional annotations, whereas both SPROF-GO and ATGO+ utilize the CAFA protocol, retaining all GO terms in experimentally verified annotations. Notably, a majority of functional annotations correspond to these less frequent GO terms, underscoring the substantial variance in training data across these methodologies. Presenting the performance of different models without aligning their training data could mislead interpretations.

2. To ascertain the practical applicability and generalizability of PhiGnet, the authors have the opportunity to evaluate its performance using the CAFA5 test data, available at this URL: <https://www.kaggle.com/competitions/cafa-5-protein-function-prediction/overview>. CAFA5, co-organized with the UniProt team, provides the benchmark for this assessment. Recently, the CAFA5 organizers finalized the results and enabled late submissions. The authors can leverage functional annotations up to August 21, 2003, to train PhiGnet and subsequently evaluate its performance on the CAFA5 test data.

Response to Q1 and Q2:

Both points 1 and 2 refer to the same point, i.e. that the different training datasets for PhiGnet and the other models precluded a direct comparison. We thank the reviewer for suggesting to use standardized datasets for training and testing. We were not able to use the CAFA5 dataset because we did not get the comments in time. We could however use the CAFA3 dataset. For fair comparison, we re-trained all machine-learning based models using their default settings, including PhiGnet, over the CAFA3 training dataset, and evaluated their performance on the CAFA3 test proteins using F-score and AUPR. The comparisons are highlighted in blue in the revised manuscript. In summary, PhiGnet retains its superiority over other models when trained on identical datasets.

Reviewers' Comments:

Reviewer #2:

Remarks to the Author:

I thank the authors for addressing my remaining concern well.

One last issue that should be fixed (at the discretion of the authors): The architecture of the FC block is not explained anywhere, as far as I could see, and neither is how the LxD-dimensional output of the GCNs are connected to the FC block with L-independent input size. Thanks.

Reviewer #3:

Remarks to the Author:

The manuscript has been improved after revisions. Here are some additional suggestions to strengthen this work.

Major

1. The authors are suggested to also incorporate the CAFA5 results of PhiGnet into the paper, which represents the latest efforts of CAFA community on protein function annotation. Although the authors have report the performance of PhiGnet over CAFA3, it was held seven years ago and can hardly represent the latest progress of automated protein function annotation.
2. As the authors wrote in line 218 (page 8), "To address homology issues, proteins sharing over 30% sequence identity with the test proteins were excluded from the training dataset". Please also show the performance of PhiGnet and all competing method over the CAFA3 test set without removing redundancy in training data. Furthermore, similar to CAFA setting, you can show the performance of different methods over difficult test proteins, which have sequence identity less than 0.6 to the proteins in the training data.
3. As wrote in line 168 (page 7), "PhiGnet slightly underperformed at sequence identity thresholds of 30% and 40%." Can you explain why it would perform worse than FunFams at lower sequence identity thresholds?
4. As mentioned in section "Statistic-informed graph networks", the authors used softmax to compute the prediction probability of each label. However, protein function prediction is a multi-label classification problem and softmax may be more suitable for multi-class classification problems. Could the authors please explain in more detail how the final predicted scores were obtained?

Minor

1. Figure 1 needs to be adjusted, e.g. the bottom of figure 1.b is obscured by figure 1.c; there is also a formatting problem with the caption in figure 1, "with orange" should not be on a separate line.

Response to Reviewer #2:

I thank the authors for addressing my remaining concern well.

One last issue that should be fixed (at the discretion of the authors): The architecture of the FC block is not explained anywhere, as far as I could see, and neither is how the $L \times D$ -dimensional output of the GCNs are connected to the FC block with L -independent input size. Thanks.

Response:

The output of the GCNs is a tensor with dimensions $L \times D$, where L represents the number of nodes in the graph. To consolidate the information across the L dimension, we apply *Sum Pooling*, effectively reducing L to 1 while preserving the other dimension. This aggregated tensor, now of size $1 \times D$, is then forwarded to the FC block (two fully connected layers, and it has been revised into "FC layers" in Fig. 1a) for further processing and predictions. We have clarified this procedure within the context highlighted in blue in the revised manuscript and Supplementary Materials.

Response to Reviewer #3:

The manuscript has been improved after revisions. Here are some additional suggestions to strengthen this work.

Major

1. The authors are suggested to also incorporate the CAFA5 results of PhiGnet into the paper, which represents the latest efforts of CAFA community on protein function annotation. Although the authors have report the performance of PhiGnet over CAFA3, it was held seven years ago and can hardly represent the latest progress of automated protein function annotation.

Response:

As explained in the previous round of review, CAFA5 was not available when this work was performed.

We will utilise the CAFA5 dataset in our future endeavors, to enable us to develop new models that align more closely with the latest advancements in the field. We are dedicated to continuously improving our method and look forward to incorporating the CAFA5 dataset to further enhance our method when it becomes available.

2. As the authors wrote in line 218 (page 8), "To address homology issues, proteins sharing over 30% sequence identity with the test proteins were excluded from the training dataset". Please also show the performance of PhiGnet and all competing method over the CAFA3 test set without removing redundancy in training data. Furthermore, similar to CAFA setting, you can show the performance of different methods over difficult test proteins, which have sequence identity less than 0.6 to the proteins in the training data.

Response:

As you suggested, we have carried out the comparisons, and please find the comparisons in Figure S12 of the Supplementary Materials.

3. As wrote in line 168 (page 7), "PhiGnet slightly underperformed at sequence identity thresholds of 30% and 40%." Can you explain why it would perform worse than FunFams at lower sequence identity thresholds?

Response:

The performance discrepancy observed at lower sequence identity thresholds between PhiGnet and FunFams can be attributed to differences in how their scores are calculated. While FunFams' prediction scores are determined based on the frequencies of function annotations of proteins, our method calculates scores by comparing predicted annotations to the ground truth.

This distinction is crucial because FunFams' scoring measurement may not fully capture the accuracy and precision of predicted annotations, especially at lower sequence identity thresholds where functional similarities between proteins might be more challenging to discern. In contrast, our approach directly evaluates the predictive accuracy of annotations against known ground truth.

Therefore, while FunFams may exhibit superior performance at certain sequence identity thresholds based on its scoring measurement, our method offers an evaluation that emphasizes the accuracy of predicted annotations relative to the ground truth.

4. As mentioned in section "Statistic-informed graph networks", the authors used softmax to compute the prediction probability of each label. However, protein function prediction is a multi-label classification problem and softmax may be more suitable for multi-class classification problems. Could the authors please explain in more detail how the final predicted scores were obtained?

Response:

The model's output is structured as a tensor with dimensions $F \times 2$, where F represents the number of classes within each function category, such as enzyme commission (EC) number, and biological processes (BP) terms. Within this tensor, the first column comprises probabilities (p_i) corresponding to annotation labels, while the second column contains the complementary probabilities ($1-p_i$). We applied *Softmax* to rescale each row of the tensor. This calculation allows for a nuanced representation of prediction certainty, with each label assigned a probability score that reflects the model's confidence in its association with the given input.

Minor

1. Figure 1 needs to be adjusted, e.g. the bottom of figure 1.b is obscured by figure 1.c; there is also a formatting problem with the caption in figure 1, "with orange" should not be on a separate line.

Response:

We have repositioned the figures in Figure 1 to enhance clarity. Please read the relocated Figure 1c along with the improved caption in the revised manuscript.